# MagNet: A Neural Network for Directed Graphs

Xitong Zhang[1], Yixuan He[2], Nathan Brugnone[1,3], Michael Perlmutter[4], and Matthew Hirn[1,5,6]

[1]Michigan State University, Department of Computational Mathematics, Science & Engineering,
East Lansing, Michigan, United States
[2]University of Oxford, Department of Statistics, Oxford, England, United Kingdom
[3]Michigan State University, Department of Community Sustainability,
East Lansing, Michigan, United States
[4]University of California, Los Angeles, Department of Mathematics,
Los Angeles, California, United States
[5]Michigan State University, Department of Mathematics,
East Lansing, Michigan, United States
[6]Michigan State University, Center for Quantum Computing, Science & Engineering,
East Lansing, Michigan, United States

## Abstract

The prevalence of graph-based data has spurred the rapid development of graph neural networks (GNNs) and related machine learning algorithms. Yet, despite the many datasets naturally modeled as directed graphs, including citation, website, and traffic networks, the vast majority of this research focuses on undirected graphs. In this paper, we propose *MagNet*, a GNN for directed graphs based on a complex Hermitian matrix known as the magnetic Laplacian. This matrix encodes undirected geometric structure in the magnitude of its entries and directional information in their phase. A "charge" parameter attunes spectral information to variation among directed cycles. We apply our network to a variety of directed graph node classification and link prediction tasks showing that MagNet performs well on all tasks and that its performance exceeds all other methods on a majority of such tasks. The underlying principles of MagNet are such that it can be adapted to other GNN architectures.

## 1 Introduction

Endowing a collection of objects with a graph structure allows one to encode pairwise relationships among its elements. These relations often possess a natural notion of direction. For example, the WebKB dataset [36] contains a list of university websites with associated hyperlinks. In this context, one website might link to a second without a reciprocal link to the first. Such datasets are naturally modeled by *directed graphs*. In this paper, we introduce *MagNet*, a graph convolutional neural network for directed graphs based on the magnetic Laplacian.

Most graph neural networks fall into one of two families, *spectral networks* or *spatial networks*. Spatial methods define graph convolution as a localized averaging operation with iteratively learned weights. Spectral networks, on the other hand, define convolution on graphs via the eigendecompositon of the (normalized) graph Laplacian. The eigenvectors of the graph Laplacian assume the role of Fourier modes, and convolution is defined as entrywise multiplication in the Fourier basis. For a comprehensive review of both spatial and spectral networks, we refer the reader to [46] and [44].

Many spatial graph CNNs have natural extensions to directed graphs. However, these extensions typically only consider the outgoing neighbors of each vertex and neglect the incoming neighbors. Therefore, they run the risk of discarding potentially important information. Consider, for example, a

35th Conference on Neural Information Processing Systems (NeurIPS 2021), virtual.

directed social network such as Twitter, where the nodes are Twitter accounts and a directed edge $(u, v) \in E$ means that account $u$ mentions account $v$ (using the @ functionality). To infer something about account $v$, there is important information to be gathered both from other accounts that $v$ mentions, and accounts that mention $v$. Therefore, it is common for spatial methods to preprocess the data by symmetrizing the adjacency matrix, effectively creating an undirected graph. For example, while [43] explicitly notes that their network is well-defined on directed graphs, their experiments treat all citation networks as undirected for improved performance.

Extending spectral methods to directed graphs is not straightforward since the adjacency matrix is asymmetric and, thus, there is no obvious way to define a symmetric, real-valued Laplacian with a full set of real eigenvalues that uniquely encodes any directed graph. We overcome this challenge by constructing a network based on the magnetic Laplacian $\mathbf{L}^{(q)}$ defined in Section 2. Unlike the directed graph Laplacians used in works such as [29, 33, 41, 42], the magnetic Laplacian is not a real-valued symmetric matrix. Instead, it is a *complex-valued Hermitian* matrix that encodes the fundamentally asymmetric nature of a directed graph via the complex phase of its entries.

Since $\mathbf{L}^{(q)}$ is Hermitian, the spectral theorem implies it has an orthonormal basis of complex eigenvectors corresponding to real eigenvalues. Moreover, Theorem 1, stated in Section 5 of the supplement, shows that $\mathbf{L}^{(q)}$ is positive semidefinite, similar to the traditional Laplacian. Setting $q = 0$ is equivalent to symmetrizing the adjacency matrix and no importance is given to directional information. When $q = .25$, on the other hand, we have that $\mathbf{L}^{(.25)}(u, v) = -\mathbf{L}^{(.25)}(v, u)$ whenever there is an edge from $u$ to $v$ but not from $v$ to $u$. Different values of $q$ highlight different graph motifs [16, 17, 20, 32], and therefore the optimal choice of $q$ varies. Learning the appropriate value of $q$ from data allows MagNet to adaptively incorporate directed information. We also note that $\mathbf{L}^{(q)}$ has been applied to graph signal processing [19], community detection [17], and clustering [10, 16, 15].

In Section 3, we show how the networks constructed in [6, 13, 24] can be adapted to directed graphs by incorporating complex Hermitian matrices, such as the magnetic Laplacian. When $q = 0$, we effectively recover the networks constructed in those previous works. Therefore, our work generalizes these networks in a way that is suitable for directed graphs. Our method is very general and is not tied to any particular choice of network architecture. Indeed, the main ideas of this work could be adapted to nearly any spectral graph neural network, and some spatial ones.

In Section 4, we summarize related work on directed graph neural networks as well as other papers studying the magnetic Laplacian and its applications in data science. In Section 5, we apply our network to node classification and link prediction tasks. We compare against several spectral and spatial methods as well as networks designed for directed graphs. We find that MagNet obtains the best or second-best performance on five out of six node-classification tasks and has the best performance on seven out of eight link-prediction tasks tested on real-world data, in addition to providing excellent node-classification performance on difficult synthetic data. We also provide a supplementary document with full implementation details, theoretical results concerning the magnetic Laplacian, extended examples, and further numerical details.

## 2   The magnetic Laplacian

Spectral graph theory has been remarkably successful in relating geometric characteristics of undirected graphs to properties of eigenvectors and eigenvalues of graph Laplacians and related matrices. For example, the tasks of optimal graph partitioning, sparsification, clustering, and embedding may be approximated by eigenvectors corresponding to small eigenvalues of various Laplacians (see, e.g., [9, 38, 2, 40, 11]). Similarly, the graph signal processing research community leverages the full set of eigenvectors to extend the Fourier transform to these structures [34]. Furthermore, numerous papers [6, 13, 24] have shown that this eigendecomposition can be used to define neural networks on graphs. In this section, we provide the background needed to extend these constructions to directed graphs via complex Hermitian matrices such as the magnetic Laplacian.

We let $G = (V, E)$ be a directed graph where $V$ is a set of $N$ vertices and $E \subseteq V \times V$ is a set of directed edges. If $(u, v) \in E$, then we say there is an edge from $u$ to $v$. For the sake of simplicity, we will focus on the case where the graph is unweighted and has no self-loops, i.e., $(v, v) \notin E$, but our methods have natural extensions to graphs with self-loops and/or weighted edges. If both $(u, v) \in E$ and $(v, u) \in E$, then one may consider this pair of directed edges as a single undirected edge.

A directed graph can be described by an adjacency matrix $(\mathbf{A}(u,v))_{u,v\in V}$ where $\mathbf{A}(u,v)=1$ if $(u,v)\in E$ and $\mathbf{A}(u,v)=0$ otherwise. Unless $G$ is undirected, $\mathbf{A}$ is not symmetric, and, indeed, this is the key technical challenge in extending spectral graph neural networks to directed graphs. In the undirected case, where the adjacency matrix $\mathbf{A}$ is symmetric, the (unnormalized) graph Laplacian can be defined by $\mathbf{L}=\mathbf{D}-\mathbf{A}$, where $\mathbf{D}$ is a diagonal degree matrix. It is well-known that $\mathbf{L}$ is a symmetric, positive-semidefinite matrix and therefore has an orthonormal basis of eigenvectors associated with non-negative eigenvalues. However, when $\mathbf{A}$ is asymmetric, direct attempts to define the Laplacian this way typically yield complex eigenvalues. This impedes the straightforward extension of classical methods of spectral graph theory and graph signal processing to directed graphs.

A key point of this paper is to represent the directed graph through a complex Hermitian matrix $\mathcal{L}$ such that: (1) the magnitude of $\mathcal{L}(u,v)$ indicates the presence of an edge, but not its direction; and (2) the phase of $\mathcal{L}(u,v)$ indicates the direction of the edge, or if the edge is undirected. Such matrices have been explored in the directed graph literature (see Section 4), but not in the context of graph neural networks. They have several advantages over their real-valued matrix counterparts. In particular, a single symmetric real-valued matrix will not uniquely represent a directed graph. Instead, one must use several matrices, as in [42], but this increases the complexity of the resulting network. Alternatively, one can work with an asymmetric, real-valued matrix, such as the adjacency matrix or the random walk matrix. However, the spatial graph filters that result from such matrices are typically limited by the fact that they can only aggregate information from the vertices that can be reached in one hop from a central vertex, but ignore the equally important subset of vertices that can reach the central vertex in one hop. Complex Hermitian matrices, however, lead to filters that aggregate information from both sets of vertices. Finally, one could use a real-valued skew-symmetric matrix but such matrices do not generalize well to graphs with both directed and undirected edges.

The optimal choice of complex Hermitian matrix is an open question. Here, we utilize a parameterized family of magnetic Laplacians, which have proven to be useful in other data-driven contexts [17, 10, 16, 15]. We first define the symmetrized adjacency matrix and corresponding degree matrix by,

$$\mathbf{A}_s(u,v) := \frac{1}{2}(\mathbf{A}(u,v)+\mathbf{A}(v,u)), \;\; 1\leq u,v\leq N, \quad \mathbf{D}_s(u,u) := \sum_{v\in V}\mathbf{A}_s(u,v), \;\; 1\leq u\leq N\,,$$

with $\mathbf{D}_s(u,v)=0$ for $u\neq v$. We capture directional information via a phase matrix,[1] $\mathbf{\Theta}^{(q)}$,

$$\mathbf{\Theta}^{(q)}(u,v) := 2\pi q(\mathbf{A}(u,v)-\mathbf{A}(v,u))\,, \quad q\geq 0\,,$$

where $\exp(i\mathbf{\Theta}^{(q)})$ is defined component-wise by $\exp(i\mathbf{\Theta}^{(q)})(u,v) := \exp(i\mathbf{\Theta}^{(q)}(u,v))$. Letting $\odot$ denote component-wise multiplication, we define the complex Hermitian adjacency matrix $\mathbf{H}^{(q)}$ by

$$\mathbf{H}^{(q)} := \mathbf{A}_s \odot \exp(i\mathbf{\Theta}^{(q)})\,.$$

Since $\mathbf{\Theta}^{(q)}$ is skew-symmetric, $\mathbf{H}^{(q)}$ is Hermitian. When $q=0$, we have $\mathbf{\Theta}^{(0)}=\mathbf{0}$ and so $\mathbf{H}^{(0)}=\mathbf{A}_s$. This effectively corresponds to treating the graph as undirected. For $q\neq 0$, the phase of $\mathbf{H}^{(q)}(u,v)$ encodes edge direction and the value $\mathbf{H}^{(q)}(u,v)$ separates four possible cases: no edge, edge from $u$ to $v$, edge from $v$ to $u$, and undirected edge. If there is no edge, we will have $\mathbf{H}^q(u,v)=0$. In the case of a directed edge, the Hermitian adjacency will be complex valued, and changing the direction of an edge will correspond to complex conjugation. For example, in the case where $q=.25$, if there is an edge from $u$ to $v$ but not from $v$ to $u$ we have

$$\mathbf{H}^{(.25)}(u,v) = \frac{i}{2} = -\mathbf{H}^{(.25)}(v,u)\,.$$

Thus, in this setting, an edge from $u$ to $v$ is treated as the opposite of an edge from $v$ to $u$. On the other hand, if $(u,v),(v,u)\in E$ (which can be interpreted as a single undirected edge), then $\mathbf{H}^{(q)}(u,v)=\mathbf{H}^{(q)}(v,u)=1$, and we see the phase, $\mathbf{\Theta}^{(q)}(u,v)=0$, encodes the lack of direction in the edge. For the rest of this paper, we will assume that $q$ lies in between these two extreme values, i.e., $0\leq q\leq .25$. We define the normalized and unnormalized magnetic Laplacians by

$$\mathbf{L}_U^{(q)} := \mathbf{D}_s - \mathbf{H}^{(q)} = \mathbf{D}_s - \mathbf{A}_s\odot\exp(i\mathbf{\Theta}^{(q)}), \quad \mathbf{L}_N^{(q)} := \mathbf{I} - \left(\mathbf{D}_s^{-1/2}\mathbf{A}_s\mathbf{D}_s^{-1/2}\right)\odot\exp(i\mathbf{\Theta}^{(q)})\,. \quad (1)$$

---

[1]Our definition of $\mathbf{\Theta}^{(q)}$ coincides with that used in [19]. However, another definition (differing by a minus sign) also appears in the literature. These resulting magnetic Laplacians have the same eigenvalues and the corresponding eigenvectors are complex conjugates of one another. Therefore, this difference does not affect the performance of our network since our final layer separates the real and imaginary parts before multiplying by a trainable weight matrix (see Section 3 for details on the network structure).

Note that when $G$ is undirected, $\mathbf{L}_U^{(q)}$ and $\mathbf{L}_N^{(q)}$ reduce to the standard undirected Laplacians.

$\mathbf{L}_U^{(q)}$ and $\mathbf{L}_N^{(q)}$ are Hermitian. Theorem 1 (Section 5 of the supplement) shows they are positive-semidefinite and thus are diagonalized by an orthonormal basis of complex eigenvectors $\mathbf{u}_1, \ldots, \mathbf{u}_N$ associated to real, nonnegative eigenvalues $\lambda_1, \ldots, \lambda_N$. Similar to the traditional normalized Laplacian, Theorem 2 (Section 5 of the supplement) shows the eigenvalues of $\mathbf{L}_N^q$ lie in $[0, 2]$, and we may factor $\mathbf{L}_N^{(q)} = \mathbf{U}\boldsymbol{\Lambda}\mathbf{U}^\dagger$, where $\mathbf{U}$ is the $N \times N$ matrix whose $k$-th column is $\mathbf{u}_k$, $\boldsymbol{\Lambda}$ is the diagonal matrix with $\boldsymbol{\Lambda}(k, k) = \lambda_k$, and $\mathbf{U}^\dagger$ is the conjugate transpose of $\mathbf{U}$ (a similar formula holds for $\mathbf{L}_U^{(q)}$). Furthermore, recall $\mathbf{L} = \mathbf{B}\mathbf{B}^\top$, where $\mathbf{B}$ is the signed incidence matrix. Similarly, Theorem 3 (Section 5 of the supplement) shows that $\mathbf{L}_U^{(q)} = \mathbf{B}^{(q)}(\mathbf{B}^{(q)})^\dagger$, where $\mathbf{B}^{(q)}$ is a modified incidence matrix. The magnetic Laplacian encodes geometric information in its eigenvectors and eigenvalues. In the directed star graph (Section 6 of the supplement), for example, directional information is contained in the eigenvectors only, whereas the eigenvalues are invariant to the direction of the edges. On the other hand, for the directed cycle graph the magnetic Laplacian encodes the directed nature of the graph solely in its spectrum. In general, both the eigenvectors and eigenvalues may contain important information, which we leverage in MagNet.

## 3  MagNet

Most graph neural network architectures can be described as being either *spectral* or *spatial*. Spatial networks such as [43, 21, 1, 14] typically extend convolution to graphs by performing a weighted average of features over neighborhoods $\mathcal{N}(u) = \{v : (u, v) \in E\}$. These neighborhoods are well-defined even when $E$ is not symmetric, so spatial methods typically have natural extensions to directed graphs. However, such simplistic extensions may miss important information in the directed graph. For example, filters defined using $\mathcal{N}(u)$ are not capable of assimilating the equally important information contained in $\{v : (v, u) \in E\}$. Alternatively, these methods may also use the symmetrized adjacency matrix, but they cannot learn to balance directed and undirected approaches.

In this section, we show how to extend spectral methods to directed graphs using the magnetic Laplacian introduced in Section 2. To highlight the flexibility of our approach, we show how three spectral graph neural network architectures can be adapted to incorporate the magnetic Laplacian. Our approach is very general, and so for most of this section, we will perform our analysis for a general complex Hermitian, positive semidefinite matrix. However, we view the magnetic Laplacian as our primary object of interest (and use it in all of our experiments in Section 5) because of the large body of literature studying its spectral properties and applying it to data science (see Section 4).

### 3.1  Spectral convolution via the magnetic Laplacian

In this section, we let $\mathcal{L}$ denote a Hermitian, positive semidefinite matrix, such as the normalized or unnormalized magnetic Laplacian introduced in Section 2, on a directed graph $G = (V, E)$, $|V| = N$. We let $\mathbf{u}_1 \ldots, \mathbf{u}_N$ be an orthonormal basis of eigenvectors for $\mathcal{L}$ and let $\mathbf{U}$ be the $N \times N$ matrix whose $k$-th column is $\mathbf{u}_k$. We define the directed graph Fourier transform for a signal $\mathbf{x} : V \to \mathbb{C}$ by $\widehat{\mathbf{x}} = \mathbf{U}^\dagger\mathbf{x}$, so that $\widehat{\mathbf{x}}(k) = \langle \mathbf{x}, \mathbf{u}_k \rangle$. We regard the eigenvectors $\mathbf{u}_1, \ldots, \mathbf{u}_N$ as the generalizations of discrete Fourier modes to directed graphs. Since $\mathbf{U}$ is unitary, we have the Fourier inversion formula

$$\mathbf{x} = \mathbf{U}\widehat{\mathbf{x}} = \sum_{k=1}^{N} \widehat{\mathbf{x}}(k)\mathbf{u}_k \, . \tag{2}$$

In Euclidean space, convolution corresponds to pointwise multiplication in the Fourier basis. Thus, we define the convolution of $\mathbf{x}$ with a filter $\mathbf{y}$ in the Fourier domain by $\widehat{\mathbf{y} * \mathbf{x}}(k) = \widehat{\mathbf{y}}(k)\widehat{\mathbf{x}}(k)$. By (2), this implies $\mathbf{y} * \mathbf{x} = \mathbf{U}\text{Diag}(\widehat{\mathbf{y}})\widehat{\mathbf{x}} = (\mathbf{U}\text{Diag}(\widehat{\mathbf{y}})\mathbf{U}^\dagger)\mathbf{x}$, and so we say $\mathbf{Y}$ is a convolution matrix if

$$\mathbf{Y} = \mathbf{U}\boldsymbol{\Sigma}\mathbf{U}^\dagger \, , \tag{3}$$

for a diagonal matrix $\boldsymbol{\Sigma}$. This is the natural generalization of the class of convolutions used in [6].

Next, following [13] (see also [22]), we show that a spectral network can be implemented in the spatial domain via polynomials of $\mathcal{L}$ by having $\boldsymbol{\Sigma}$ be a polynomial of $\boldsymbol{\Lambda}$ in (3). This reduces the number of trainable parameters to prevent overfitting, avoids explicit diagonalization of the matrix $\mathcal{L}$,

(which is expensive for large graphs), and improves stability to perturbations [27]. As in [13], we define a normalized eigenvalue matrix, with entries in $[-1, 1]$, by $\widetilde{\mathbf{\Lambda}} = \frac{2}{\lambda_{\max}}\mathbf{\Lambda} - \mathbf{I}$ and assume

$$\mathbf{\Sigma} = \sum_{k=0}^{K} \theta_k T_k(\widetilde{\mathbf{\Lambda}}),$$

for some real-valued $\theta_1, \ldots, \theta_k$, where $T_k$ is the Chebyshev polynomial defined by $T_0(x) = 1, T_1(x) = x$, and $T_k(x) = 2xT_{k-1}(x) + T_{k-2}(x)$ for $k \geq 2$. With $(\mathbf{U}\widetilde{\mathbf{\Lambda}}\mathbf{U}^\dagger)^k = \mathbf{U}\widetilde{\mathbf{\Lambda}}^k\mathbf{U}^\dagger$, one has

$$\mathbf{Y}\mathbf{x} = \mathbf{U}\sum_{k=0}^{K}\theta_k T_k(\widetilde{\mathbf{\Lambda}})\mathbf{U}^\dagger\mathbf{x} = \sum_{k=0}^{K}\theta_k T_k(\widetilde{\mathcal{L}})\mathbf{x}, \tag{4}$$

where, analogous to $\widetilde{\mathbf{\Lambda}}$, we define $\widetilde{\mathcal{L}} := \frac{2}{\lambda_{\max}}\mathcal{L} - \mathbf{I}$. It is important to note that, due to the complex Hermitian structure of $\widetilde{\mathcal{L}}$, the value $\mathbf{Y}\mathbf{x}(u)$ aggregates information both from the values of $\mathbf{x}$ on $\mathcal{N}_k(u)$, the $k$-hop neighborhood of $u$, and the values of $\mathbf{x}$ on $\{v : \text{dist}(v, u) \leq k\}$, which consists of those of vertices that can reach $u$ in $k$-hops. While in an undirected graph these two sets of vertices are the same, that is not the case for general directed graphs. Furthermore, due to the difference in phase between an edge $(u, v)$ and an edge $(v, u)$, the filter matrix $\mathbf{Y}$ is also capable of aggregating information from these two sets in different ways. This capability is in contrast to any single, symmetric, real-valued matrix, as well as any matrix that encodes just $\mathcal{N}(u)$.

To obtain a network similar to [24], we set $K = 1$, assume that $\mathcal{L} = \mathbf{L}_N^{(q)}$, using $\lambda_{\max} \leq 2$ make the approximation $\lambda_{\max} \approx 2$, and set $\theta_1 = -\theta_0$. With this, we obtain

$$\mathbf{Y}\mathbf{x} = \theta_0(\mathbf{I} + (\mathbf{D}_s^{-1/2}\mathbf{A}_s\mathbf{D}_s^{-1/2}) \odot \exp(i\mathbf{\Theta}^{(q)}))\mathbf{x}.$$

As in [24], we substitute $\mathbf{I} + (\mathbf{D}_s^{-1/2}\mathbf{A}_s\mathbf{D}_s^{-1/2}) \odot \exp(i\mathbf{\Theta}^{(q)}) \to \widetilde{\mathbf{D}}_s^{-1/2}\widetilde{\mathbf{A}}_s\widetilde{\mathbf{D}}_s^{-1/2} \odot \exp(i\mathbf{\Theta}^{(q)})$. This renormalization helps avoid instabilities arising from vanishing/exploding gradients and yields

$$\mathbf{Y}\mathbf{x} = \theta_0\widetilde{\mathbf{D}}_s^{-1/2}\widetilde{\mathbf{A}}_s\widetilde{\mathbf{D}}_s^{-1/2} \odot \exp(i\mathbf{\Theta}^{(q)}), \tag{5}$$

where $\widetilde{\mathbf{A}}_s = \mathbf{A}_s + \mathbf{I}$ and $\widetilde{\mathbf{D}}_s(i, i) = \sum_j \widetilde{\mathbf{A}}_s(i, j)$.

In theory, the matrix $\exp(i\mathbf{\Theta}^{(q)})$ is dense. However, in practice, one only needs to compute a small fraction of its entries. In most real-world datasets, the symmetrized adjacency matrix will be sparse. Since the Hermitian adjacency matrix is constructed via pointwise multiplication between the symmetrized adjacency matrix and the phase matrix, it is only necessary to compute the phase matrix for entries $(u, v)$ where $\mathbf{A}_s(u, v) \neq 0$. Thus, the efficiency of the proposed algorithm is comparable to standard GCN algorithms, and can leverage any existing developments such as [18] that increase efficiency of standard GCNs (although the computational complexity our method does differ by a factor of four because of the computational complexity of complex-valued multiplication).

## 3.2 The MagNet architecture

Let $L$ be the number of convolution layers in our network, and let $\mathbf{X}^{(0)}$ be an $N \times F_0$ input feature matrix with columns $\mathbf{x}_1^{(0)}, \ldots \mathbf{x}_{F_0}^{(0)}$. Since our filters are complex, we use a complex version of ReLU defined by $\sigma(z) = z$, if $-\pi/2 \leq \arg(z) < \pi/2$, and $\sigma(z) = 0$ otherwise (where $\arg(z)$ is the complex argument of $z \in \mathbb{C}$). Let $F_\ell$ be the number of channels in layer $\ell$, and for $1 \leq \ell \leq L$, $1 \leq i \leq F_{\ell-1}$, and $1 \leq j \leq F_\ell$, we let $\mathbf{Y}_{ij}^{(\ell)}$ be a convolution matrix defined in the sense of either (3), (4), or (5). Define the $\ell^{\text{th}}$ layer feature matrix $\mathbf{X}^{(\ell)}$ with columns $\mathbf{x}_1^{(\ell)}, \ldots \mathbf{x}_{F_\ell}^{(\ell)}$ as:

$$\mathbf{x}_j^{(\ell)} = \sigma\left(\sum_{i=1}^{F_{\ell-1}} \mathbf{Y}_{ij}^{(\ell)}\mathbf{x}_i^{(\ell-1)} + \mathbf{b}_j^{(\ell)}\right), \tag{6}$$

with $\mathbf{b}_j^{(\ell)}(v) = b_j^{(\ell)}$ and $\text{real}(b_j^{(\ell)}) = \text{imag}(b_j^{(\ell)})$. In matrix form we write $\mathbf{X}^{(\ell)} = \mathbf{Z}^{(\ell)}\left(\mathbf{X}^{(\ell-1)}\right)$, where $\mathbf{Z}^{(\ell)}$ is a hidden layer of the form (6). In the numerical experiments reported in Section 5, we utilize formulation (4) with $\mathcal{L} = \mathbf{L}_N^{(q)}$. In most cases we set $K = 1$, for which

$$\mathbf{X}^{(\ell)} = \sigma\left(\mathbf{X}^{(\ell-1)}\mathbf{W}_{\text{self}}^{(\ell)} + \widetilde{\mathbf{L}}_N^{(q)}\mathbf{X}^{(\ell-1)}\mathbf{W}_{\text{neigh}}^{(\ell)} + \mathbf{B}^{(\ell)}\right),$$

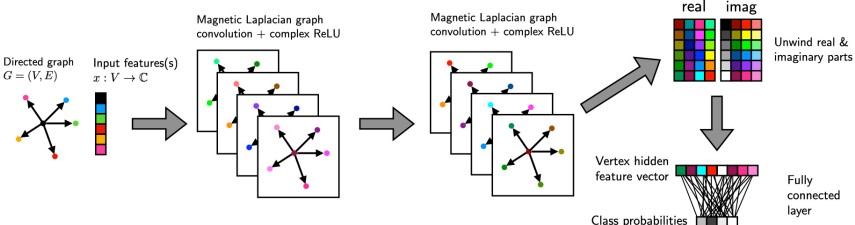

Figure 1: MagNet ($L = 2$) applied to node classification.

where $\mathbf{W}_{\text{self}}^{(\ell)}$ and $\mathbf{W}_{\text{neigh}}^{(\ell)}$ are learned weight matrices corresponding to the filter weights in (4), and $\mathbf{B}^{(\ell)}(v, \cdot) = (b_1^{(\ell)}, \ldots, b_{F_\ell}^{(\ell)})$ for each $v \in V$.

After the convolutional layers, we unwind the complex $N \times F_L$ matrix $\mathbf{X}^{(L)}$ into a real-valued $N \times 2F_L$ matrix, apply a linear layer, consisting of right-multiplication by a $2F_L \times n_c$ weight matrix $\mathbf{W}^{(L+1)}$ (where $n_c$ is the number of classes) and apply softmax. In our experiments, we set $L = 2$ or 3. When $L = 2$, our network applied to node classification, as illustrated in Figure 1, is given by

$$\text{softmax}(\text{unwind}(\mathbf{Z}^{(2)}(\mathbf{Z}^{(1)}(\mathbf{X}^{(0)})))) \mathbf{W}^{(3)}) \,.$$

For link-prediction, we apply the same method through the unwind layer, and then concatenate the rows corresponding to pairs of nodes to obtain the edge features.

## 4 Related work

In Section 4.1, we describe other graph neural networks designed specifically for directed graphs. Notably, none of these methods encode directionality with complex numbers, instead opting for real-valued, symmetric matrices. In Section 4.2, we review other work studying the magnetic Laplacian which has been studied for several decades and lately has garnered interest in the network science and graph signal processing communities. However, to the best of our knowledge, this is the first work to use it to construct a graph neural network. We also note there are numerous approaches to graph signal processing on directed graphs. Many of these rely on a natural analog of Fourier modes. These Fourier modes are typically defined through either a factorization of a graph shift operator or by solving an optimization problem. For further review, we refer the reader to [30].

### 4.1 Neural networks for directed graphs

In [29], the authors construct a directed Laplacian, via identities involving the random walk matrix and its stationary distribution $\mathbf{\Pi}$. When $G$ is undirected, one can use the fact that $\mathbf{\Pi}$ is proportional to the degree vector to verify this directed Laplacian reduces to the standard normalized graph Laplacian. However, this method requires $G$ to be strongly connected, unlike MagNet. The authors of [42] use a first-order proximity matrix $\mathbf{A}_F$ (equivalent to $\mathbf{A}_s$ here), as well as two second-order proximity matrices $\mathbf{A}_{S_{\text{in}}}$ and $\mathbf{A}_{S_{\text{out}}}$. $\mathbf{A}_{S_{\text{in}}}$ is defined by $\mathbf{A}_{S_{\text{in}}}(u, v) \neq 0$ if there exists a $w$ such that $(w, u), (w, v) \in E$, and $\mathbf{A}_{S_{\text{out}}}$ is defined analogously. These three matrices collectively describe and distinguish the neighborhood of each vertex and those vertices that can reach a vertex in a single hop. The authors construct three different Laplacians and use a fusion operator to share information across channels. Similarly, inspired by [3], in [33], the authors consider several different symmetric Laplacian matrices corresponding to a number of different graph motifs.

The method of [41] builds upon the ideas of both [29] and [42] and considers a directed Laplacian similar to the one used in [29], but with a PageRank matrix in place of the random-walk matrix. This allows for applications to graphs which are not strongly connected. Similar to [42], they use higher-order receptive fields (analogous to the second-order adjacency matrices discussed above) and an inception module to share information between receptive fields of different orders. We also note [25], which uses an approach based on PageRank in the spatial domain. There are also some related methods for directed graphs that are not based on the graph Laplacian, such as the directed graph embedding [39], and directed message passing for molecular graphs [26].

## 4.2 Related work on the magnetic Laplacian and Hermitian adjacency matrices

The magnetic Laplacian has been studied since at least [28]. The name originates from its interpretation as a quantum mechanical Hamiltonian of a particle under magnetic flux. Early works focused on $d$-regular graphs, where the eigenvectors of the magnetic Laplacian are equivalent to those of the Hermitian adjacency matrix. The authors of [20], for example, show that using a complex-valued Hermitian adjacency matrix rather than the symmetrized adjacency matrix reduces the number of small, non-isomorphic cospectral graphs. Topics of current research into Hermitian adjacency matrices include clustering tasks [12] and the role of the parameter $q$ [32].

The magnetic Laplacian is also the subject of ongoing research in graph signal processing [19], community detection [17], and clustering [10, 16, 15]. For example, [16] uses the phase of the eigenvectors to construct eigenmap embeddings analogous to [2]. The role of $q$ is highlighted in the works of [16, 17, 20, 32], which show how particular choices of $q$ may highlight various graph motifs. In our context, this indicates that $q$ should be carefully tuned via cross-validation. Lastly, we note that numerous other directed graph Laplacians have been studied and applied to data science [7, 8, 35]. However, as alluded to in Section 2, these methods typically do not use complex Hermitian matrices.

# 5 Numerical experiments

## 5.1 Datasets

### 5.1.1 Directed Stochastic Block Model

We construct a directed stochastic block (DSBM) model as follows. First we divide $N$ vertices into $n_c$ equally-sized clusters $C_1, \ldots, C_{n_c}$. We define $\{\alpha_{i,j}\}_{1 \leq i,j \leq n_c}$ to be a collection of probabilities, $0 < \alpha_{i,j} \leq 1$ with $\alpha_{i,j} = \alpha_{j,i}$, and for an unordered pair $u \neq v$ create an undirected edge between $u$ and $v$ with probability $\alpha_{i,j}$ if $u \in C_i, v \in C_j$. To turn this undirected graph into a directed graph, we define $\{\beta_{i,j}\}_{1 \leq i,j \leq n_c}$ to be a collection of probabilities such that $0 \leq \beta_{i,j} \leq 1$ and $\beta_{i,j} + \beta_{j,i} = 1$. For each undirected edge $\{u, v\}$, we assign that edge a direction by the rule that the edge points from $u$ to $v$ with probability $\beta_{i,j}$ if $u \in C_i$ and $v \in C_j$, and points from $v$ to $u$ otherwise. If $\alpha_{i,j}$ is constant, then the only way to determine the clusters will be from the directional information.

In Figure 3, we plot the performance of MagNet and other methods on variations of the DSBM. In each of these, we set $n_c = 5$ and the goal is to classify the vertices by cluster. We set $N = 2500$, except in Figure 3d where $N = 500$. In Figure 3a, we plot the performance of our model on the DSBM with $\alpha_{i,j} := \alpha^* = .1, .08$, and .05 for $i \neq j$, which varies the density of inter-cluster edges, and set $\alpha_{i,i} = .1$. Here we set $\beta_{i,i} = .5$ and $\beta_{i,j} = .05$ for $i > j$. This corresponds to the ordered meta-graph in Figure 2a. Figure 3b also uses the ordered meta-graph, but here we fix $\alpha_{i,j} = .1$ for all $i, j$, and set $\beta_{i,j} = \beta^*$, for $i > j$, and allow $\beta^*$ to vary from .05 to .4, which varies the net flow (related to flow imbalance in [23]) from one cluster to another. The results in Figure 3c utilize a cyclic meta-graph structure as in Figure 2b (without the gray noise edges). Specifically, we set $\alpha_{i,j} = .1$ if $i = j$ or $i = j \pm 1 \mod 5$ and $\alpha_{i,j} = 0$ otherwise. We define $\beta_{i,j} = \beta^*, \beta_{j,i} = 1 - \beta^*$ when $j = (i - 1) \mod 5$, and $\beta_{i,j} = 0$ otherwise. In Figure 3d we add

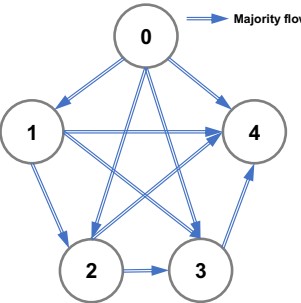

(a) Ordered meta-graph.

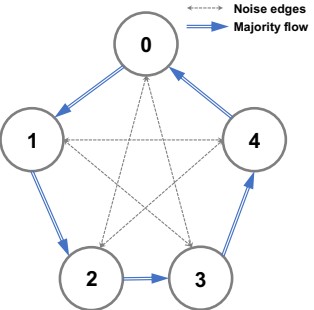

(b) Cyclic meta-graph.

Figure 2: Meta-graphs for the synthetic data sets.

noise to the cyclic structure of our meta-graph by setting $\alpha_{i,j} = .1$ for all $i, j$ and $\beta_{i,j} = .5$ for all $(i, j)$ connected by a gray edge in Figure 2b (keeping $\beta_{i,j}$ the same as in Figure 3c for the blue edges).

### 5.1.2 Real datasets

*Texas*, *Wisconsin*, and *Cornell* are WebKB datasets modeling links between websites at different universities [36]. We use these datasets for both link prediction and node classification with nodes

labeled as student, project, course, staff, and faculty in the latter case. *Telegram* [5] is a pairwise influence network between 245 Telegram channels with $8,912$ links. To the best of our knowledge, this dataset has not previously been studied in the graph neural network literature. Labels are generated from the method discussed in [5], with a total of four classes. The datasets *Chameleon* and *Squirrel* [37] represent links between Wikipedia pages related to chameleons and squirrels. We use these datasets for link prediction. Likewise, *WikiCS* [31] is a collection of Computer Science articles, which we also use for link prediction (see the tables in Section 7 of the supplement). *Cora-ML* and *CiteSeer* are popular citation networks with node labels corresponding to scientific subareas. We use the versions of these datasets provided in [4]. Further details are given in the supplementary material.

## 5.2 Training and implementation details

Node classification is performed in a semi-supervised setting (i.e., access to the test data, but not the test labels, during training). For the datasets *Cornell*, *Texas*, *Wisconsin*, and *Telegram* we use a 60%/20%/20% training/validation/test split, which might be viewed as more akin to supervised learning, because of the small graph size. For *Cora-ML* and *CiteSeer*, we use the same split as [41]. For all of these datasets we use 10 random data splits. For the DSBM datasets, we generated 5 graphs randomly for each type and for each set of parameters, each with 10 different random node splits. We use $20\%$ of the nodes for validation and we vary the proportion of training samples based on the classification difficulty, using 2%, 10%, and 60% of nodes per class for the ordered, cyclic, and noisy cyclic DSBM graphs, respectively, during training, and the rest for testing. Hyperpameters were selected using one of the five generated graphs, and then applied to the other four generated graphs.

In the main text, there are two types of link prediction tasks conducted for performance evaluation. The first type is to predict the edge direction of pairs of vertices $u, v$ for which either $(u, v) \in E$ or $(v, u) \in E$. The second type is existence prediction. The model is asked to predict if $(u, v) \in E$ by considering ordered pairs of vertices $(u, v)$. For both types of link prediction, we removed 15% of edges for testing, 5% for validation, and use the rest of the edges for training. The connectivity was maintained during splitting. 10 splits were generated randomly for each graph and the input features are in-degree and out-degree of nodes. In the supplement, we report on two additional link prediction tasks based on a three-class classification setup: $(u, v) \in E, (v, u) \in E$, or $(u, v), (v, u) \notin E$. Full details are provided in the supplementary material.

In all experiments, we used the normalized magnetic Laplacian and implement MagNet with convolution defined as in (4), meaning that our network may be viewed as the magnetic Laplacian generalization of ChebNet. The setting of the hyperparameter $q$ and other network hyperparameters is obtained by cross-validation. Since currently complex tensors are still in beta in PyTorch, we did not use them, and instead we stored any complex tensor as two real tensors (one for the real part, one for the imaginary part), and carried out complex multiplication using the standard formula: $(a + ib)(c + id) = (ac - bd) + i(bc + ad)$ (note, $a, b, c, d$ can be real numbers or real matrices). We compare with multiple baselines in three categories: (i) spectral methods: ChebNet [13], GCN [24]; (ii) spatial methods: APPNP [25], SAGE [21], GIN [45], GAT [43]; and (iii) methods designed for directed graphs: DGCN [42], and two variants of [41], a basic version (DiGraph) and a version with higher order inception blocks (DiGraphIB). All baselines in the experiments have two graph convolutional layers, except for the node classification on the DSBM using the cyclic meta-graphs (Figures 3c, 3d, and 2b) for which we also tested three layers during the hyperparameter search. For ChebNet, we use the symmetrized adjacency matrix. For the spatial networks we apply both the symmetrized and asymmmetric adjacency matrix for node classification. The results reported are the better of the two results. The supplement provides full details, as well as results for two other types of baselines: (i) BiGCN, BiSAGE, BiGAT which are obtained by applying GCN, SAGE, GAT on both the original adjacency matrix and the transposed adjacency matrix; and (ii) a $k$-nearest neighbors classifier based on the eigenvector with the smallest eigenvalue of the magnetic Laplacian [17].

## 5.3 Results

We see that MagNet performs well across all tasks. As indicated in Table 1, our cross-validation procedure selects $q = 0$ for node classification on the citation networks *Cora-ML* and *CiteSeer*. This means we achieved the best performance when regarding directional information as noise, suggesting symmetrization-based methods are appropriate in the context of node classification on citation networks. This matches our intuition. For example, in *Cora-ML*, the task is to classify

Table 1: Node classification accuracy (%). The best results are in **bold** and the second are underlined.

| Type | Method | Cornell | Texas | Wisconsin | Cora-ML | CiteSeer | Telegram | Score |
|------|--------|---------|-------|-----------|---------|----------|----------|-------|
| Spectral | ChebNet | 79.8±5.0 | 79.2±7.5 | 81.6±6.3 | 80.0±1.8 | 66.7±1.6 | 70.2 ±6.8 | 6.94 |
| | GCN | 59.0±6.4 | 58.7±3.8 | 55.9±5.4 | 82.0±1.1 | 66.0±1.5 | 73.4 ±5.8 | 19.16 |
| Spatial | APPNP | 58.7±4.0 | 57.0±4.8 | 51.8±7.4 | **82.6±1.4** | 66.9±1.8 | 67.3±3.0 | 18.75 |
| | SAGE | 80.0±6.1 | **84.3±5.5** | 83.1±4.8 | 82.3±1.2 | 66.0±1.5 | 66.4±6.4 | 5.76 |
| | GIN | 57.9±5.7 | 65.2±6.5 | 58.2±5.1 | 78.1±2.0 | 63.3±2.5 | 86.4±4.3 | 16.53 |
| | GAT | 57.6±4.9 | 61.1±5.0 | 54.1±4.2 | 81.9±1.0 | 67.3±1.3 | 72.6±7.5 | 16.39 |
| Directed | DGCN | 67.3±4.3 | 71.7±7.4 | 65.5±4.7 | 81.3±1.4 | 66.3±2.0 | **90.4±5.6** | 8.55 |
| | Digraph | 66.8±6.2 | 64.9±8.1 | 59.6±3.8 | 79.4±1.8 | 62.6±2.2 | 82.0±3.1 | 15.70 |
| | DiGraphIB | 64.4±9.0 | 64.9±13.7 | 64.1±7.0 | 79.3±1.2 | 61.1±1.7 | 64.1±7.0 | 16.36 |
| This paper | MagNet | **84.3±7.0** | 83.3±6.1 | **85.7±3.2** | 79.8±2.5 | **67.5±1.8** | 87.6±2.9 | **1.10** |
| | Best $q$ | 0.25 | 0.15 | 0.05 | 0.0 | 0.0 | 0.15 | - |

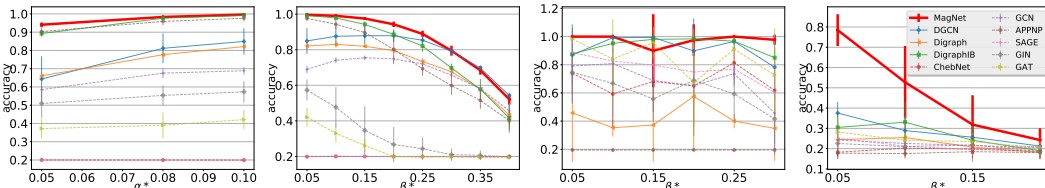

(a) Ordered DSBM with varying edge density.  (b) Ordered DSBM with varying net flow.  (c) Cyclic DSBM with varying net flow.  (d) Noisy Cyclic DSBM with varying net flow.

Figure 3: Node classification accuracy. Error bars are one standard error. MagNet is bold red.

research papers by scientific subarea. If the topic of a given paper is "machine learning," then it is likely to both cite and be cited by other machine learning papers. For all other datasets, we find the optimal value of $q$ is nonzero, indicating that directional information is important. Our network exhibits the best performance on three out of six of these datasets and is a close second on *Texas* and *Telegram*. We also achieve an at least four percentage point improvement over both ChebNet and GCN on the four data sets for which $q > 0$. These networks are similar to ours but with the classical graph Laplacian. This isolates the effects of the magnetic Laplacian and shows that it is a valuable tool for encoding directional information. MagNet also compares favorably to non-spectral methods on the WebKB networks (*Cornell*, *Texas*, *Wisconsin*). Indeed, MagNet obtains a $\sim 4\%$ improvement on *Cornell* and a $\sim 2.5\%$ improvement on *Wisconsin*, while on *Texas* it has the second best accuracy, close behind SAGE. We also see the other directed methods have relatively poor performance on the WebKB networks, perhaps since these graphs are fairly small and have very few training samples. To make this analysis more quantitative, we computed the absolute difference of the classification accuracy of each method from the classification accuracy of the top performing method (in percentage points) on each data set, and averaged over the six data sets. In this context, lower scores are better, and a method with a score of zero indicates the method is the top performing method on each data set. As reported in Table 1, MagNet achieved a best score of 1.1 percent.

On the DSBM datasets, as illustrated in Figure 3 (see also the tables in Section 7 of the supplement), we see that MagNet generally performs quite well and is the best performing network in the vast majority of cases (for full details, see Section 7 of the supplement). The networks DGCN and DiGraphIB rely on second order proximity matrices. As demonstrated in Figure 3c, these methods are well suited for networks with a cyclic meta-graph structure since nodes in the same cluster are likely to have common neighbors. MagNet, on the other hand, does not use second-order proximity, but can still learn the clusters by stacking multiple layers together. This improves MagNet's ability to adapt to directed graphs with different underlying topologies. This is illustrated in Figure 3d where the network has an approximately cyclic meta-graph structure. In this setting, MagNet continues to perform well, but the performance of DGCN and DiGraphIB deteriorate significantly. Interestingly, MagNet performs well on the DSBM cyclic meta-graph (Figure 3c) with $q \approx .1$, whereas $q \geq .2$ is

Table 2: Link prediction accuracy (%). The best results are in **bold** and the second are underlined.

| | Direction prediction | | | | Existence prediction | | | |
|---|---|---|---|---|---|---|---|---|
| | Cornell | Wisconsin | Cora-ML | CiteSeer | Cornell | Wisconsin | Cora-ML | CiteSeer |
| ChebNet | 71.0±5.5 | 67.5±4.5 | 72.7±1.5 | 68.0±1.6 | 80.1±2.3 | 82.5±1.9 | 80.0±0.6 | 77.4±0.4 |
| GCN | 56.2±8.7 | 71.0±4.0 | 79.8±1.1 | 68.9±2.8 | 75.1±1.4 | 75.1±1.9 | 81.6±0.5 | 76.9±0.5 |
| APPNP | 69.5±9.0 | 75.1±3.5 | 83.7±0.7 | 77.9±1.6 | 74.9±1.5 | 75.7±2.2 | 82.5±0.6 | 78.6±0.7 |
| SAGE | 75.2±11.0 | 72.0±3.5 | 68.2±0.8 | 68.7±1.5 | 79.8±2.4 | 77.3±2.9 | 75.0±0.0 | 74.1±1.0 |
| GIN | 69.3±6.0 | 74.8±3.7 | 83.2±0.9 | 76.3±1.4 | 74.5±2.1 | 76.2±1.9 | 82.5±0.7 | 77.9±0.7 |
| GAT | 67.9±11.1 | 53.2±2.6 | 50.0±0.1 | 50.6±0.5 | 77.9±3.2 | 74.6±0.0 | 75.0±0.0 | 75.0±0.0 |
| DGCN | 80.7±6.3 | 74.5±7.2 | 79.6±1.5 | 78.5±2.3 | 80.0±3.9 | **82.8±2.0** | 82.1±0.5 | 81.2±0.4 |
| DiGraph | 79.3±1.9 | 82.3±4.9 | 80.8±1.1 | 81.0±1.1 | 80.6±2.5 | **82.8±2.6** | 81.8±0.5 | **82.2±0.6** |
| DiGraphIB | 79.8±4.8 | 82.0±4.9 | 83.4±1.1 | 82.5±1.3 | 80.5±3.6 | 82.4±2.2 | 82.2±0.5 | 81.0±0.5 |
| MagNet | **82.9±3.5** | **83.3±3.0** | **86.5±0.7** | **84.8±1.2** | **81.1±3.3** | **82.8±2.2** | **82.7±0.7** | 79.9±0.6 |
| Best $q$ | 0.20 | 0.10 | 0.20 | 0.15 | 0.25 | 0.05 | 0.05 | 0.05 |

preferred for the other three DSBM tests; we leave a more in-depth investigation for future work. Further details are available in Section 8 of the supplement.

For link prediction, we achieve the best performance on seven out of eight tests as shown in Table 2. We also note that Table 2 reports optimal non-zero $q$ values for each task. This indicates that incorporating directional information is important for link prediction, even on citation networks such as *Cora* and *CiteSeer*. This matches our intuition, since there is a clear difference between a paper with many citations and one with many references. More results on different datasets, and closely related tasks (including three-class classification), are provided in Section 7 in the supplement.

## 6    Conclusion

We have introduced MagNet, a neural network for directed graphs based on the magnetic Laplacian. This network can be viewed as the natural extension of spectral graph convolutional networks to the directed graph setting. We demonstrate the effectiveness of our network, and the importance of incorporating directional information via a complex Hermitian matrix, for link prediction and node classification on both real and synthetic datasets. Interesting avenues of future research would be using multiple $q$'s along different channels, exploring the role of different normalizations of the magnetic Laplacian, and incorporating the magnetic Laplacian into other network architectures.

**Limitations and Ethical Considerations:** Our method has natural extensions to weighted, directed graphs when all edges are directed. However, it not clear what is the best way to extend it to weighted mixed graphs (with both directed and undirected edges). Our network does not incorporate an attention mechanism and, similar to many other networks, is not scalable to large graphs in its current form (although this may be addressed in future work). All of our data is publicly available for research purposes and does not contain personally identifiable information or offensive content. The method presented here has no greater or lesser impact on society than other graph neural network algorithms.

## Acknowledgments

We would like to thank Jie Zhang who pointed out that our definition of the magnetic Laplacian differed by an entry-wise complex conjugation from the most commonly used definition in the literature. Y.H. thanks her supervisors Mihai Cucuringu and Gesine Reinert for their guidance.

This work was supported by the National Institutes of Health [grant NIGMS-R01GM135929 to M.H. and supporting X.Z, N.B.]; the University of Oxford [the Clarendon scholarship to Y.H.]; the National Science Foundation [grant DMS-1845856 to M.H.]; and the Department of Energy [grant DE-SC0021152 to M.H.].

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
