# MagNet: A Neural Network for Directed Graphs
# Supplementary Material

Xitong Zhang[1], Yixuan He[2], Nathan Brugnone[1,3], Michael Perlmutter[4], and Matthew Hirn[1,5,6]

[1]Michigan State University, Department of Computational Mathematics, Science & Engineering,
East Lansing, Michigan, United States
[2]University of Oxford, Department of Statistics, Oxford, England, United Kingdom
[3]Michigan State University, Department of Community Sustainability,
East Lansing, Michigan, United States
[4]University of California, Los Angeles, Department of Mathematics,
Los Angeles, California, United States
[5]Michigan State University, Department of Mathematics,
East Lansing, Michigan, United States
[6]Michigan State University, Center for Quantum Computing, Science & Engineering,
East Lansing, Michigan, United States

## Contents

35th Conference on Neural Information Processing Systems (NeurIPS 2021), virtual.

# 1  Github repository

A Github repository containing code needed to reproduce the results is https://github.com/matthew-hirn/magnet.

# 2  List of method abbreviations

- MagNet (this paper)
- ChebNet [1]
- GCN [5]
- APPNP [6]
- GAT [10]
- SAGE [4]
- GIN [11]
- DGCN [9]
- DiGraph [8]
- DiGraphIB [8]: DiGraph with inception blocks

- BiGCN: applying GCN on the original adjacency matrix and its transpose matrix separately
- BiSAGE: applying SAGE on the original adjacency matrix and its transpose matrix separately
- BiGAT: applying GAT on the original adjacency matrix and its transpose matrix separately
- KNN: K-nearest neighbors based on the eigenvectors with the smallest eigenvalues of magnetic Laplacian [3].

# 3  Further implementation details

We set the parameter $K = 1$ in our implementation of both ChebNet and MagNet, except for synthetic noisy cylcic graphs with random input features. For sythetic noisy cylcic graphs with random input features, we also tried $K = 2$ for MagNet. We train all models with a maximum of 3000 epochs and stop early if the validation error doesn't decrease after $500$ epochs for both node classification and link prediction tasks. One dropout layer with a probability of $0.5$ is created before the last linear layer. The model is picked with the best validation accuracy during training for testing. We tune the number of filters in $[16, 32, 48]$ for the graph convolutional layers for all models, except DigraphIB, since the inception block has more trainable parameters. For node classification, we tune the learning rate in $[1e^{-3}, 5e^{-3}, 1e^{-2}]$ for all models. Compared with node classification, the number of available samples for link prediction is much larger. Thus, we set a relatively small learning rate of $1e^{-3}$.

We use Adam as the optimizer and $\ell_2$ regularization with the hyperparameter set as $5e^{-4}$ to avoid overfitting. We post the best testing performance by grid-searching based on validation accuracy. For node classification on the synthetic datasets, we generate a one-dimensional node feature sampled from the standard normal distribution. We use the original features for the other node classification datasets. For link prediction, we use the in-degree and out-degree as the node features for all datasets instead the original features. This allows all models to learn directed information from the adjacency matrix. Our experiments were conducted on 8 compute nodes each with 1 Nvidia Tesla V100 GPU, 120G RAM, and 32 Intel Xeon E5-2660 v3 CPUs; as well as on a compute node with 8 Nvidia RTX 8000 GPUs, 1000GB RAM, and 48 Intel Xeon Silver 4116 CPUs.

Here are implementation details specific to certain methods:

- We set the parameter $\epsilon$ to 0 in GIN for both tasks.
- For GAT and BiGAT, the number of heads tuned is in $[2, 4, 8]$.
- For APPNP, we set $K = 10$ for node classification (following the original paper [6]), and search K in $[1, 5, 10]$ for link prediction.
- The coefficient $\alpha$ for PageRank-based models (APPNP, DiGraph) is searched in $[0.05, 0.1, 0.15, 0.2]$.
- For DiGraph, the model includes graph convolutional layers without the high-order approximation and inception module. The high order Laplacian and the inception module is included in DigraphIB.
- DigraphIB is a bit different than other networks because it requires generating a three-channel Laplacian tensor. For this network, the number of filters for each channel is searched in $[6, 11, 21]$ for node classification and link prediction.

- For GCN, the out-degree normalized, directed adjacency matrix, including self-loops is also tried in addition to the symmetrized adjacency matrix for node classification tasks, except for synthetic datasets since symmetrization will break the cluster pattern.
- For other spatial methods, including APPNP, GAT, SAGE, and GIN, we tried both the symmetrized adjacency matrices and the original directed adjacency matrices for node classification tasks except for synthetic datasets.
- For KNN, we set $q = 0.25$ and $K = 5$.

## 4  Datasets

### 4.1  Node classification

As shown in Table 1, we use six real datasets for node classification. A directed edge is defined as follows. If the edge $(u, v) \in E$ but $(v, u) \notin E$, then $(u, v)$ is a directed edge. If $(u, v) \in E$ and $(v, u) \in E$, then $(u, v)$ and $(v, u)$ are undirected edges (in other words, undirected edges that are not self-loops are counted twice). For the citation datasets, *Cora-ML* and *Citeseer*, we randomly select 20 nodes in each class for training, 500 nodes for validation, and the rest for testing following [8]. For the synthetic datasets (*ordered DSBM graphs*, *cyclic DSBM graphs*, *noisy cyclic DSBM graphs*), we generate a one-dimensional node feature sampled from the standard normal distribution.

Ten folds are generated randomly for each dataset, except for *Cornell*, *Texas* and *Wisconsin*. For *Cornell*, *Texas*, and *Wisconsin*, we use the same training, validation, and testing folds as [7]. For *Telegram*, we treat it as a directed, unweighted graph and randomly generate 10 splits for training/validation/testing with 60%/20%/20% of the nodes. The node features are sampled from the normal distribution.

Table 1: Real datasets for node classification.

|                   | Cornell | Texas | Wisconsin | Cora-ML | Citeseer | Telegram |
|-------------------|---------|-------|-----------|---------|----------|----------|
| # Nodes           | 183     | 183   | 251       | 2,995   | 3,312    | 245      |
| # Edges           | 295     | 309   | 499       | 8,416   | 4,715    | 8,912    |
| % Directed edges  | 86.9    | 76.6  | 77.9      | 93.9    | 95.0     | 82.4     |
| # Features        | 1,703   | 1,703 | 1,703     | 2,879   | 3,703    | 1        |
| # Classes         | 5       | 5     | 5         | 7       | 6        | 4        |

### 4.2  Link prediction

We use eight real datasets in link prediction as demonstrated in Table 2. Instead of using the original features, we use the in-degree and out-degree as the node features in order to allow the models to learn structural information from the adjacency matrix directly. The connectivity is maintained by getting the undirected minimum spanning tree before removing edges for validation and testing. For the results in the main text, undirected edges and, if they exist, pairs of vertices with multiple edges between them, may be placed in the training/validation/testing sets. However, labels that indicate the direction of such edges are not well defined, and therefore can be considered as noisy labels from the machine learning perspective. In order to obtain a full set of well-defined, noiseless labels, in the supplement we also run experiments in which undirected edges and pairs of vertices with multiple edges between them are ignored when sampling edges for training/validation/testing (in other words, only directed edges, and the absence of an edge, are included). We evaluated all models on four prediction tasks, which we now describe.

To construct the datasets that we use for training, validation and testing, which consist of pairs of vertices in the graph, we do the following. (1) Existence prediction. If $(u, v) \in E$, we give $(u, v)$ the label 0, otherwise its label is 1. The proportion of the two classes of edges is 25% and 75%, respectively, when undirected edges and multi-edges are included, and 50% and 50%, respectively, when only directed edges are included. (2) Direction prediction. Given an ordered node pair $(u, v)$, we give the label 0 if $(u, v) \in E$ and the label 1 if $(v, u) \in E$, conditioning on $(u, v) \in E$ or $(v, u) \in E$. The proportion of the two types of edges is 50% and 50%. (3) Three-class link prediction. For a pair of ordered nodes $(u, v)$, if $(u, v) \in E$, we give the label 0, if $(v, u) \in E$, we give the

label 1, if $(u, v) \notin E$ and $(v, u) \notin E$, we give the label 2. The proportion of the three types of edges is 25%, 25%, and 50%. (4) Direction prediction by three classes training. This task is based on the training of task (3). We only evaluate the performance with ordered node pairs $(u, v)$ when $(u, v) \in E$ or $(v, u) \in E$. We randomly generated ten folds for all datasets. We used 15% and 5% of edges for testing and validation for all datasets. The classification results are in Section 7.

Table 2: Datasets for link prediction.

|  | Cornell | Texas | Wisconsin | Cora-ML | CiteSeer | WikiCS | Chameleon | Squirrel |
|---|---|---|---|---|---|---|---|---|
| # Nodes | 183 | 183 | 251 | 2,995 | 3,312 | 11,701 | 2,277 | 5,201 |
| # Edges | 295 | 309 | 499 | 8,416 | 4,715 | 216,123 | 36,101 | 217,073 |
| % Directed edges | 86.9 | 76.6 | 77.9 | 93.9 | 95.0 | 45.9 | 73.9 | 82.8 |
| # Features | 2 | 2 | 2 | 2 | 2 | 2 | 2 | 2 |

## 5  Eigenvalues of the magnetic Laplacian

In this section we state and prove three theorems. Theorem 1, which shows that both the normalized and unnormalized magnetic Laplacian a postive semidefinite, is well known (see e.g. [2]). Theorem 2, which shows that the eigenvalues of the normalized magnetic Laplacian lie in the interval $[0, 2]$, is a straightforward adaption of the corresponding result for the traditional normalized graph Laplacian. Finally, Theorem 3 proves the un-normalized magnetic Laplacian may be factored in terms of a complex valued incidence matrix, analogous to the well-known result for the standard graph Laplacian. We give full proofs of all three results for completeness.

**Theorem 1.** *Let $G = (V, E)$ be a directed graph where $V$ is a set of $N$ vertices and $E \subseteq V \times V$ is a set of directed edges. Then, for all $q \geq 0$, both the unnormalized magnetic Laplacian $\mathbf{L}_U^{(q)}$ and its normalized counterpart $\mathbf{L}_N^{(q)}$ are positive semidefinite.*

*Proof.* Let $\mathbf{x} \in \mathbb{C}^N$. We first note that since $\mathbf{L}_U^{(q)}$ is Hermitian we have $\mathrm{Imag}(\mathbf{x}^\dagger \mathbf{L}_U^{(q)} \mathbf{x}) = 0$. Next, we use the definition of $\mathbf{D}_s$ and the fact that $\mathbf{A}_s$ is symmetric to observe that

$$
2\mathrm{Real}\left(\mathbf{x}^\dagger \mathbf{L}_U^{(q)} \mathbf{x}\right)
$$

$$
= 2\sum_{u,v=1}^{N} \mathbf{D}_s(u,v)\mathbf{x}(u)\overline{\mathbf{x}(v)} - 2\sum_{u,v=1}^{N} \mathbf{A}_s(u,v)\mathbf{x}(u)\overline{\mathbf{x}(v)}\cos(i\mathbf{\Theta}^{(q)}(u,v))
$$

$$
= 2\sum_{u=1}^{N} \mathbf{D}_s(u,u)\mathbf{x}(u)\overline{\mathbf{x}(u)} - 2\sum_{u,v=1}^{N} \mathbf{A}_s(u,v)\mathbf{x}(u)\overline{\mathbf{x}(v)}\cos(i\mathbf{\Theta}^{(q)}(u,v))
$$

$$
= 2\sum_{u,v=1}^{N} \mathbf{A}_s(u,v)|\mathbf{x}(u)|^2 - 2\sum_{u,v=1}^{N} \mathbf{A}_s(u,v)\mathbf{x}(u)\overline{\mathbf{x}(v)}\cos(i\mathbf{\Theta}^{(q)}(u,v))
$$

$$
= \sum_{u,v=1}^{N} \mathbf{A}_s(u,v)|\mathbf{x}(u)|^2 + \sum_{u,v=1}^{N} \mathbf{A}_s(v,u)|\mathbf{x}(v)|^2 - 2\sum_{u,v=1}^{N} \mathbf{A}_s(u,v)\mathbf{x}(u)\overline{\mathbf{x}(v)}\cos(i\mathbf{\Theta}^{(q)}(u,v))
$$

$$
= \sum_{u,v=1}^{N} \mathbf{A}_s(u,v)\left(|\mathbf{x}(u)|^2 + |\mathbf{x}(v)|^2 - 2\mathbf{x}(u)\overline{\mathbf{x}(v)}\cos(i\mathbf{\Theta}^{(q)}(u,v))\right) \tag{1}
$$

$$
\geq \sum_{u,v=1}^{N} \mathbf{A}_s(u,v)\left(|\mathbf{x}(u)|^2 + |\mathbf{x}(v)|^2 - 2|\mathbf{x}(u)||\mathbf{x}(v)|\right)
$$

$$
= \sum_{u,v=1}^{N} \mathbf{A}_s(u,v)(|\mathbf{x}(u)| - |\mathbf{x}(v)|)^2
$$

$$
\geq 0.
$$

Thus, $\mathbf{L}_U^{(q)}$ is positive semidefinite. For the normalized magnetic Laplacian, we note that

$$\left(\mathbf{D}_s^{-1/2}\mathbf{A}_s\mathbf{D}_s^{-1/2}\right) \odot \exp(i\Theta^{(q)}) = \mathbf{D}_s^{-1/2}\left(\mathbf{A}_s \odot \exp(i\Theta^{(q)})\right)\mathbf{D}_s^{-1/2},$$

and therefore

$$\mathbf{L}_N^{(q)} = \mathbf{D}_s^{-1/2}\mathbf{L}_U^{(q)}\mathbf{D}_s^{-1/2}. \tag{2}$$

Thus, letting $\mathbf{y} = \mathbf{D}_s^{-1/2}\mathbf{x}$, the fact that $\mathbf{D}_s$ is diagonal implies

$$\mathbf{x}^\dagger\mathbf{L}_N^{(q)}\mathbf{x} = \mathbf{x}^\dagger\mathbf{D}_s^{-1/2}\mathbf{L}_U^{(q)}\mathbf{D}_s^{-1/2}\mathbf{x} = \mathbf{y}^\dagger\mathbf{L}_U^{(q)}\mathbf{y} \geq 0.$$

$\square$

**Theorem 2.** *Let $G = (V, E)$ be a directed graph where $V$ is a set of $N$ vertices and $E \subseteq V \times V$ is a set of directed edges. Then, for all $q \geq 0$, the eigenvalues of the normalized magnetic Laplacian $\mathbf{L}_N^{(q)}$ are contained in the interval $[0, 2]$.*

*Proof.* By Theorem 1, we know that $\mathbf{L}_N^{(q)}$ has real, nonnegative eigenvalues. Therefore, we need to show that the lead eigenvalue, $\lambda_N$, is less than or equal to 2. The Courant-Fischer theorem shows that

$$\lambda_N = \max_{\mathbf{x}\neq 0} \frac{\mathbf{x}^\dagger\mathbf{L}_N^{(q)}\mathbf{x}}{\mathbf{x}^\dagger\mathbf{x}}.$$

Therefore, using (2) and setting $\mathbf{y} = \mathbf{D}_s^{-1/2}\mathbf{x}$, we have

$$\lambda_N = \max_{\mathbf{x}\neq 0} \frac{\mathbf{x}^\dagger\mathbf{D}_s^{-1/2}\mathbf{L}_U^{(q)}\mathbf{D}_s^{-1/2}\mathbf{x}}{\mathbf{x}^\dagger\mathbf{x}} = \max_{\mathbf{y}\neq 0} \frac{\mathbf{y}^\dagger\mathbf{L}_U^{(q)}\mathbf{y}}{\mathbf{y}^\dagger\mathbf{D}_s\mathbf{y}}.$$

First, we observe that since $\mathbf{D}_s$ is diagonal, we have

$$\mathbf{y}^\dagger\mathbf{D}_s\mathbf{y} = \sum_{u,v=1}^{N} \mathbf{D}_s(u,v)\mathbf{y}(u)\overline{\mathbf{y}(v)} = \sum_{u=1}^{N} \mathbf{D}_s(u,u)|\mathbf{y}(u)|^2$$

Next, we note that by (1), we have

$$\mathbf{y}^\dagger\mathbf{L}_U^{(q)}\mathbf{y} = \frac{1}{2}\sum_{u,v=1}^{N} \mathbf{A}_s(u,v)\left(|\mathbf{x}(u)|^2 + |\mathbf{x}(v)|^2 - 2\mathbf{x}(u)\overline{\mathbf{x}(v)}\cos(i\Theta^{(q)}(u,v))\right)$$

$$\leq \frac{1}{2}\sum_{u,v=1}^{N} \mathbf{A}_s(u,v)(|\mathbf{x}(u)| + |\mathbf{x}(v)|)^2$$

$$\leq \sum_{u,v=1}^{N} \mathbf{A}_s(u,v)(|\mathbf{x}(u)|^2 + |\mathbf{x}(v)|^2).$$

Therefore, since $\mathbf{A_s}$ is symmetric, we have

$$\mathbf{y}^\dagger\mathbf{L}_U^{(q)}\mathbf{y} \leq 2\sum_{u,v=1}^{N} \mathbf{A}_s(u,v)|\mathbf{x}(u)|^2$$

$$= 2\sum_{u=1}^{N} |\mathbf{x}(u)|^2 \left(\sum_{v=1}^{N} \mathbf{A}_s(u,v)\right)$$

$$= 2\sum_{u=1}^{N} \mathbf{D}_s(u,u)|\mathbf{x}(u)|^2$$

$$= 2\mathbf{y}^\dagger\mathbf{D}_s\mathbf{y}.$$

$\square$

**Definition 1.** *Let $G = (V, E)$ be a directed graph where $V$ is a set of $N$ vertices and $E \subseteq V \times V$ is a set of directed edges. We say that a link $(u, v) \in E$ is bidirectional if the "reverse" link $(v, u)$ is also also in E. If a link is not bidirectional we say that it is unidirectional.*

**Theorem 3.** *Let $G = (V, E)$ be a directed graph where $V$ is a set of $N$ vertices and $E \subseteq V \times V$ is a set of directed edges. Then, for all $q \geq 0$, the unnormalized magnetic Laplacian may be factored as $\mathbf{L}_U^{(q)} = \mathbf{B}^{(q)} (\mathbf{B}^{(q)})^\dagger$, where $\mathbf{B}^{(q)}$ is a modified incidence matrix defined by*

$$\mathbf{B}^{(q)}(j, \ell) = \begin{cases} \frac{1}{\sqrt{2}} e^{i\pi q} & \text{if } j \text{ is the source of link } \ell \text{ and } \ell \text{ is unidirectional} \\ \frac{-1}{\sqrt{2}} e^{-i\pi q} & \text{if } j \text{ is the sink of the link } \ell \text{ and } \ell \text{ is unidirectional} \\ 1 & \text{if } j \text{ is the source of the link } \ell \text{ and } \ell \text{ is bidirectional} \\ 1 & \text{if } j \text{ is the sink of the link } \ell \text{ and } \ell \text{ is bidirectional} \\ 0 & \text{otherwise} \end{cases} \cdot$$

*Proof.* Let $\mathbf{B} = \mathbf{B}^{(q)}$ for the remainder of the proof. By definition we have,

$$(\mathbf{B}\mathbf{B}^\dagger)(j, k) = \sum_\ell \mathbf{B}(j, \ell)\overline{\mathbf{B}(k, \ell)}$$

If $j = k$, we have

$$(\mathbf{B}\mathbf{B}^\dagger)(j, j)$$
$$= \sum_\ell \mathbf{B}(j, \ell)\overline{\mathbf{B}(j, \ell)}$$
$$= \sum_{\substack{\ell \text{ unidirectional} \\ \text{st. } j \text{ is a source}}} \mathbf{B}(j, \ell)\overline{\mathbf{B}(j, \ell)} + \sum_{\substack{\ell \text{ unidirectional} \\ \text{st. } j \text{ is a sink}}} \mathbf{B}(j, \ell)\overline{\mathbf{B}(j, \ell)} + \sum_{\substack{\ell \text{ bidirectional} \\ \text{st. } j \text{ is a source}}} \mathbf{B}(j, \ell)\overline{\mathbf{B}(j, \ell)} + \sum_{\substack{\ell \text{ bidirectional} \\ \text{st. } j \text{ is a sink}}} \mathbf{B}(j, \ell)\overline{\mathbf{B}(j, \ell)}$$
$$= \sum_{\substack{\ell \text{ unidirectional} \\ \text{st. } j \text{ is a source}}} \frac{1}{\sqrt{2}} e^{i\pi q} \overline{\frac{1}{\sqrt{2}} e^{i\pi q}} + \sum_{\substack{\ell \text{ unidirectional} \\ \text{st. } j \text{ is a sink}}} \frac{-1}{\sqrt{2}} e^{-i\pi q} \left(\overline{\frac{-1}{\sqrt{2}} e^{i\pi q}}\right) + \sum_{\substack{\ell \text{ bidirectional} \\ \text{st. } j \text{ is a source}}} 1 + \sum_{\substack{\ell \text{ bidirectional} \\ \text{st. } j \text{ is a sink}}} 1$$
$$= \frac{1}{2}(d_{in}(j) + d_{out}(j))$$
$$= d_s(j).$$

If $j \neq k$ and there is a link from $j$ to $k$ but not from $k$ to $j$, then

$$(\mathbf{B}\mathbf{B}^\dagger)(j, k) = \sum_\ell \mathbf{B}(j, \ell)\overline{\mathbf{B}(k, \ell)} = \frac{1}{\sqrt{2}} e^{i\pi q} \left(\overline{\frac{-1}{\sqrt{2}} e^{i\pi q}}\right) = \frac{-1}{2} e^{2\pi i q} = -\mathbf{H}^{(q)}(j, k)$$

Likewise, if there is a link from $k$ to $j$ but not from $j$ to $k$ we have

$$(\mathbf{B}\mathbf{B}^\dagger)(j, k) = \sum_\ell \mathbf{B}(j, \ell)\overline{\mathbf{B}(k, \ell)} = \left(\frac{-1}{\sqrt{2}} e^{-i\pi q}\right) \overline{\frac{1}{\sqrt{2}} e^{-i\pi q}} = \frac{-1}{2} e^{-2\pi i q} = -\mathbf{H}^{(q)}(j, k).$$

Lastly, if there is neither a link from $k$ to $j$ or $j$ to $k$ we have $(\mathbf{B}\mathbf{B}^\dagger)(j, k) = 0$. □

## 6 The eigenvectors and eigenvalues of directed stars and cycles

In this section, we examine the eigenvectors and eigenvalues of the unnormalized magnetic Laplacian on two example graphs. As alluded to in the main text, in the directed star graph directional information is contained in the eigenvectors only. For the directed cycle, on the other hand, the magnetic Laplacian encodes the directed nature of the graph only through the eigenvalues. Both examples can be verified via direct pen and paper calculation.

**Example 1.** *Let $G^{(in)}$ and $G^{(out)}$ be the directed star graphs with vertices $V = \{1, \dots, N\}$ and edges pointing in/out to the central vertex as shown in Figure 1. Then the eigenvalues of $\mathbf{L}_U^{(q,in)}$, the unnormalized magnetic Laplacian on $G^{in}$, are given by*

$$\lambda_1^{in} = 0, \quad \lambda_k^{in} = \frac{1}{2} \text{ for } 2 \leq k \leq N-1, \quad \text{and} \quad \lambda_N^{in} = \frac{N}{2}.$$

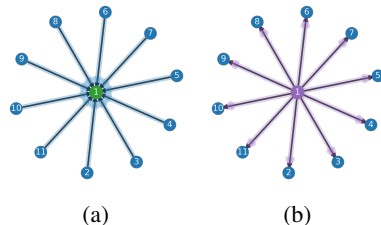

Figure 1: Directed stars (a) $G^{(\text{in})}$, and (b) $G^{(\text{out})}$

*If we let $v = 1$ be the central vertex, then the lead eigenvector is given by*

$$\mathbf{u}_1^{in}(1) = e^{2\pi iq}, \quad \mathbf{u}_1^{in}(n) = 1, \ 2 \leq n \leq N.$$

*For $2 \leq k \leq N - 1$, the eigenvectors are*

$$\mathbf{u}_k^{in} = \boldsymbol{\delta}_k - \boldsymbol{\delta}_{k+1},$$

*and the final eigenvector is given by*

$$\mathbf{u}_N^{in}(1) = -e^{2\pi iq}, \quad \mathbf{u}_N^{in}(n) = \frac{1}{N-1}, \ 2 \leq n \leq N.$$

*The phase matrices satisfies $\boldsymbol{\Theta}^{(q,in)} = -\boldsymbol{\Theta}^{(q,out)}$. Therefore, the associated magnetic Laplacians satisfy $\mathbf{L}_U^{(q,in)}(u,v) = \overline{\mathbf{L}_U^{(q,out)}(u,v)}$. Since these matrices are Hermitian, this implies that the corresponding eigenvalue-eigenvector pairs satisfy $\lambda_k^{in} = \lambda_k^{out}$, and $\mathbf{u}_k^{in} = \overline{\mathbf{u}_k^{out}}$. Hence, $\mathbf{u}_1^{in}$ and $\mathbf{u}_1^{out}$ identify the central vertex, and the sign of their imaginary parts at this vertex identifies whether it is a source or a sink. On the other hand, the eigenvalues give no directional information.*

**Example 2.** *Let $G$ be the directed cycle. Then, then the eigenvalues of $\mathbf{L}_U^{(q)}$ is are the classical Fourier modes $\mathbf{u}_k(n) = e^{(2\pi ikn/N)}$, independent of $q$. The eigenvalues, however, do depend on $q$ and are given by*

$$\lambda_k = 1 - \cos\left(2\pi\left(\frac{k}{N} + q\right)\right), \quad 1 \leq k \leq N.$$

## 7 Expanded details of numerical results

Here we present more details on our node classification results in Tables 3, 4, 5, 6, 7 and 8; and more details of our link prediction results in Tables 9, 10, 11, 12, 13, and 14. We present our results in the form mean $\pm$ standard deviation.

The networks GCN, APPNP, GAT, SAGE, and GIN were not designed with directed graphs as the primary motivation. Therefore, we implemented these methods in two ways: (i) with the original asymmetric adjacency matrix; and (ii) with a symmetrized adjacency matrix. For node classification, symmetrizing the adjacency matrix improved performance for most of these networks on most of the real datasets. We did not test the symmetric implementations on our synthetic DSBM datasets because these datasets, by design, place a heavy importance on directional information. For link prediction, on the other hand, we only use asymmetric adjacency matrices. In our tables below, GCN, APPNP, GAT, SAGE, and GIN refer to the implementations with the symmetrized adjacency matrix and GCN-D, APPNP-D, GAT-D, SAGE-D, and GIN-D refer to our implementation with the asymmetric matrix.

Tables 3, 4, 5, and 6 provide the precise node classification results for the four types of DSBM graphs introduced in Section 5.1.1 of the main text; they correspond to the plots in Figure 3 of the main text. Table 8 contains all of the information contained in Table 1 from the main text, but reports separately the results of GCN, APPNP, GAT, SAGE, and GIN (which use the symmetrized adjacency matrix) and the results of GCN-D, APPNP-D, GAT-D, SAGE-D, and GIN-D (which use the asymmetric adjacency matrix), whereas Table 1 in the main text reported only the best-performer between the two variants.

In Table 6, the spectral KNN classifier performs extremely well and exceeds the performance of MagNet. This is perhaps unsurprising since it is known that spectral clustering style methods generally perform well on stochastic block models. However, we note that in our main experiments with MagNet, we used random noise for our initial node features. If we rerun these experiments using the lead eigenvector of the magnetic Laplacian for the initial node features, then the performance of MagNet improves significantly and exceeds that of spectral clustering as shown in Table 7. For real datasets, spectral clustering performs poorly. The poor performance of spectral clustering may be caused by the inability of spectral clustering to leverage the important information which is contained in the node features of these datasets.

In Table 6, BiGCN, BiSAGE, and BiGAT are generally better than GCN-D, SAGE-D, and GAT-D. It indicates that it is beneficial to consider the neighbors in both directions. However, there is no noticeable improvement for BiGCN and BiGAT on real datasets WebKB/Cornell and Citeseer, as reported in Table 8. It is because the best performance of GCN and GAT is achieved when symmetrizing the adjacency matrix first. BiSAGE is better than SAGE on WebKB/Cornell since testing accuracy is better for SAGE with the original asymmetric adjacency matrix. But with all node classification results, MagNet outperforms BiGCN, BiSAGE, and BiGAT.

With respect to link prediction, there are many results in the supplement in addition to what is reported in the main text. Table 9 in the supplement is the same as Table 2 from the main text, except in the supplement we also include the *Texas* data set. Table 10 expands upon Table 9 by considering the more difficult three-class classification problems described in Section 4.2 of the supplement. All of the results in these tables include undirected edges and, if present, multi-edges, which have essentially random labels with respect to their directionality (see also Section 4.2), and hence these results indicate the model's ability to ignore these noisy labels. MagNet performs quite well across this slate of link prediction experiments (top performer in 13/20 experiments).

Tables 11, 12, 13, and 14 evaluate the same four link prediction tasks as Tables 9 and 10, except that undirected edges and multi-edges are not included in the training/validation/testing sets. Thus all labels are well-defined and noiseless. In this setting MagNet also performs very well, obtaining the top performance in 23/32 experiments across all four tables.

Aside from Table 13, MagNet achieves the highest testing accuracy in 20/24 experiments. Digraph achieves the highest testing accuracy in 4/8 experiments, and MagNet is best in 3/8 experiments as shown in Table 13. Having said that, there is not a statistically significant difference between MagNet and the top performing method in two other datasets (*Wisconsin* and *Cora-ML*), and MagNet is also a very close second on *WikiCS*. Thus, MagNet is either the top performer or on par with the top performing method in 5/8 datasets in Table 13. Nevertheless, the task is more difficult for MagNet than other tasks. We hypothesize that this is because half of the task is identifying whether there is an edge between $u, v$, or not; the other half, if there is an edge, is determining its direction. The first half of the task is an undirected task, and thus $q > 0$ could provide noisy features for those pairs of vertices for which there is no edge. The Digraph method utilizes the symmetric Laplacian, which is unsuitable for direction prediction but works well for predicting the presence of an edge in either direction or the absence of an edge. The direction of the edge is more important in Tables 11, 12, and 14, and MagNet captures the direction information very well. The results indicate there is a trade-off between capturing undirected and directed features. This observation also leads to a potential future research direction that utilizes magnetic Laplacian matrices based on multiple values of $q$, making MagNet capture both undirected and directed information precisely.

Tables 11, 12, and 14 also report the testing accuracy of BiGCN, BiSAGE and BiGAT. There is a significant improvement for BiGCN and BiSAGE compared with SAGE and GCN for WebKB/Cornell and Citeseer. However, BiGAT is better than GAT only on WebKB/Cornell since GAT fails on Citeseer for link prediction tasks, and there are no benefits to consider neighbors from both directions.

Table 3: Node classification accuracy of ordered DSBM graphs with varying edge density.

| Method / $\alpha^*$ | 0.1 | 0.08 | 0.05 |
|---|---|---|---|
| ChebNet | 19.9±0.6 | 20.0±0.7 | 20.0±0.7 |
| GCN-D | 68.9±2.1 | 67.6±2.7 | 58.5±2.0 |
| APPNP-D | 97.7±1.7 | 95.9±2.2 | 90.3±2.4 |
| SAGE-D | 20.1±1.1 | 19.9±0.8 | 19.9±1.0 |
| GIN-D | 57.3±5.8 | 55.4±5.5 | 50.9±7.7 |
| GAT-D | 42.1±5.3 | 39.0±7.0 | 37.2±5.5 |
| DGCN | 84.9±7.2 | 81.2±8.2 | 64.4±12.4 |
| DiGraph | 82.1±1.7 | 77.7±1.6 | 66.1±2.4 |
| DiGraphIB | 99.2±0.5 | 97.7±0.7 | 89.3±1.7 |
| MagNet | **99.6±0.2** | **98.3±0.8** | **94.1±1.2** |
| Best $q$ | 0.25 | 0.10 | 0.25 |

Table 4: Node classification accuracy of ordered DSBM graphs with varying net flow.

| Method / $\beta^*$ | .05 | .10 | .15 | .20 | .25 | .30 | .35 | .40 |
|---|---|---|---|---|---|---|---|---|
| ChebNet | 19.9±0.6 | 20.0±0.6 | 20.0±0.6 | 20.0±0.8 | 20.0±0.8 | 20.0±0.6 | 19.7±0.8 | 19.9±0.6 |
| GCN-D | 68.9±2.1 | 74.1±1.7 | 75.4±1.3 | 74.9±1.3 | 72.0±1.3 | 65.7±1.6 | 58.1±2.2 | 45.7±4.2 |
| APPNP-D | 97.7±1.7 | 94.3±2.5 | 89.6±3.5 | 80.2±8.3 | 69.3±4.0 | 59.8±5.2 | 51.5±4.6 | 40.1±5.1 |
| SAGE-D | 20.1±1.1 | 20.1±1.0 | 19.9±0.8 | 19.9±0.7 | 19.6±0.8 | 19.9±0.8 | 20.1±0.9 | 19.9±0.8 |
| GIN-D | 57.3±5.8 | 47.7±11.4 | 34.7±13.5 | 26.7±9.8 | 24.4±6.3 | 21.0±3.5 | 20.6±2.7 | 19.8±0.6 |
| GAT-D | 42.1±5.3 | 33.0±5.1 | 26.2±3.7 | 19.9±1.4 | 20.0±1.2 | 19.9±0.8 | 19.5±0.2 | 19.5±0.2 |
| DGCN | 84.9±7.2 | 87.5±5.6 | 87.9±4.8 | 87.9±3.5 | 85.3±2.5 | 79.2±2.0 | **69.7±1.5** | **54.2±1.7** |
| DiGraph | 82.1±1.7 | 83.0±1.7 | 81.9±1.1 | 79.5±1.3 | 73.4±1.8 | 67.6±2.6 | 57.9±1.5 | 43.4±6.4 |
| DiGraphIB | 99.2±0.5 | 97.9±0.7 | 94.2±1.6 | 88.6±2.2 | 82.0±3.1 | 69.7±2.5 | 57.7±5.8 | 41.6±8.2 |
| MagNet | **99.6±0.2** | **99.0±0.9** | **97.5±0.7** | **94.3±1.5** | **88.9±1.8** | **79.4±2.8** | 68.4±2.5 | 52.1±3.0 |
| Best $q$ | 0.25 | 0.20 | 0.20 | 0.25 | 0.20 | 0.20 | 0.20 | 0.25 |

Table 5: Node classification accuracy of cyclic DSBM graphs with varying net flow.

| Method / $\beta^*$ | .05 | .10 | .15 | .20 | .25 | .30 |
|---|---|---|---|---|---|---|
| ChebNet | 75.3±16.9 | 63.3±24.1 | 70.6±20.2 | 71.3±29.5 | 86.6±7.4 | 59.7±21.3 |
| GCN-D | 83.5±24.8 | 64.9±35.8 | 69.8±9.4 | 58.4±37.1 | 76.7±7.5 | 39.5±22.2 |
| APPNP-D | 19.5±0.4 | 19.6±0.5 | 19.4±0.3 | 19.6±0.7 | 19.8±0.6 | 20.1±1.5 |
| SAGE-D | 91.5±5.4 | 81.2±18.9 | 79.0±8.4 | 71.1±24.3 | 75.7±7.4 | 46.0±24.4 |
| GIN-D | 77.2±19.0 | 68.6±21.1 | 54.2±16.3 | 67.9±20.8 | 55.5±19.4 | 39.3±20.5 |
| GAT-D | 98.2±2.1 | 91.2±20.4 | 97.0±2.9 | 58.7±39.2 | 93.1±4.6 | 60.2±38.3 |
| DGCN | 91.4±15.6 | 97.9±11.7 | 99.0±1.0 | 80.2±32.9 | 95.8±6.7 | 96.0±4.7 |
| DiGraph | 71.5±30.3 | 76.9±27.4 | 73.1±33.3 | 74.8±35.1 | 85.7±12.1 | 66.3±28.4 |
| DiGraphIB | 88.0±16.5 | 97.4±2.5 | **99.4±0.5** | **98.1±1.2** | 90.8±13.4 | 88.4±6.8 |
| MagNet | **100.0±0.1** | **100.0±0.0** | 99.2±1.1 | 96.6±12.6 | **99.9±0.3** | **98.8±1.0** |
| Best $q$ | 0.10 | 0.10 | 0.15 | 0.10 | 0.10 | 0.10 |

Table 6: Node classification accuracy of noisy cyclic DSBM graphs with varying net flow.

| Method / $\beta^*$ | .05 | .10 | .15 | .20 |
|---|---|---|---|---|
| ChebNet | 18.2±3.3 | 20.2±3.5 | 19.8±3.6 | 18.6±3.9 |
| GCN-D | 24.3±4.9 | 22.6±4.5 | 21.6±3.8 | 20.5±4.8 |
| APPNP-D | 17.8±2.3 | 17.6±2.0 | 18.5±2.8 | 18.0±2.1 |
| SAGE-D | 25.2±7.9 | 21.2±4.6 | 19.6±3.3 | 21.2±4.1 |
| GIN-D | 22.6±5.1 | 21.0±4.8 | 21.8±3.3 | 18.6±2.7 |
| GAT-D | 28.2±7.6 | 24.2±4.7 | 23.5±3.9 | 20.5±4.1 |
| BiGCN | 40.6±11.2 | 25.2±5.7 | 26.4±7.6 | 17.7±2.7 |
| BiSAGE | 32.2±10.7 | 22.0±5.3 | 21.4±5.8 | 20.7±4.2 |
| BiGAT | 51.9±11.4 | 28.0±6.8 | 30.6±11.1 | 23.3±5.2 |
| DGCN | 37.6±5.5 | 28.9±6.6 | 25.6±6.8 | 21.3±3.7 |
| DiGraph | 24.4±8.3 | 25.5±10.5 | 20.8±7.4 | 18.1±3.1 |
| DiGraphIB | 30.5±7.6 | 33.0±10.1 | 24.2±9.1 | 19.0±3.7 |
| MagNet | **78.4±7.8** | **52.8±17.9** | **31.8±14.5** | **24.2±5.8** |
| Best $q$ | 0.25 | 0.25 | 0.25 | 0.25 |

Table 7: Node classification accuracy of noisy cyclic DSBM graphs with varying net flow with input features as the eigenvector of magnetic Laplacian with smallest eigenvalues.

| Method / $\beta^*$ | .05 | .10 | .15 | .20 |
|---|---|---|---|---|
| ChebNet | 91.8±2.5 | 87.4±3.5 | 81.2±4.2 | 69.9±4.3 |
| GCN-D | 63.9±10.3 | 49.8±7.8 | 41.5±6.2 | 35.9±4.3 |
| APPNP-D | 86.5±3.8 | 84.3±3.2 | 77.1±4.3 | 67.1±4.5 |
| SAGE-D | 95.1±2.4 | 90.4±3.7 | 83.8±3.1 | 70.8±4.0 |
| GIN-D | 50.4±7.8 | 43.0±8.1 | 39.3±4.5 | 35.5±4.4 |
| GAT-D | 93.1±3.3 | 87.4±4.1 | 80.3±4.5 | 64.6±4.8 |
| DGCN | 85.6±4.4 | 77.1±5.6 | 68.0±4.9 | 58.2±3.8 |
| DiGraph | 57.4±20.8 | 51.7±18.1 | 50.8±16.4 | 42.6±10.3 |
| DiGraphIB | 77.5±6.1 | 68.8±7.2 | 66.0±6.7 | 51.6±5.9 |
| KNN | 94.1±2.0 | 88.4±2.9 | 80.9±4.2 | 67.7±4.3 |
| MagNet | **97.1±1.5** | **94.1±2.2** | **88.8±3.2** | **75.1±4.3** |
| Best $q$ | 0.25 | 0.25 | 0.25 | 0.25 |

Table 8: Testing accuracy of node classification. The best results are in **bold** and the second best are underlined.

|  | Cornell | Texas | Wisconsin | Cora-ML | Citeseer | Telegram |
|---|---|---|---|---|---|---|
| ChebNet | 79.8±5.0 | 79.2±7.5 | 81.6±6.3 | 80.0±1.8 | 66.7±1.6 | 73.4 ±5.8 |
| GCN | 59.0±6.4 | 57.9±5.4 | 55.9±5.4 | 82.0±1.1 | 66.0±1.5 | 73.4±5.9 |
| GCN-D | 57.3±4.8 | 58.7±3.8 | 52.7±5.4 | 72.6±1.6 | 60.5±1.6 | 63.6±4.7 |
| APPNP | 58.7±4.0 | 57.0±4.8 | 49.6±6.5 | **82.6±1.4** | 66.9±1.8 | 69.4±3.5 |
| APPNP-D | 58.4±3.0 | 56.8±2.7 | 51.8±7.4 | 68.6±2.5 | 58.6±1.8 | 66.4±5.0 |
| GAT | 57.6±4.9 | 61.1±5.0 | 54.1±4.2 | 81.9±1.0 | 67.3±1.3 | 72.6±7.5 |
| GAT-D | 57.3±7.7 | 59.2±4.1 | 52.0±4.6 | 73.1±1.6 | 62.7±1.6 | 67.4±4.4 |
| SAGE | 77.6±6.3 | **84.3±5.5** | 79.2±5.3 | 82.3±1.2 | 66.0±1.5 | 66.4±6.4 |
| SAGE-D | 80.0±6.1 | 76.2±3.8 | 83.1±4.8 | 72.0±2.1 | 61.8±2.0 | 58.2±3.6 |
| GIN | 57.9±5.7 | 65.2±6.5 | 58.2±5.1 | 78.1±2.0 | 63.3±2.5 | 86.4±4.3 |
| GIN-D | 55.4±5.2 | 58.1±5.3 | 50.2±7.6 | 67.0±3.2 | 60.4±2.3 | 67.0±4.3 |
| BiGCN | 58.4±2.5 | - | - | - | 65.5±1.8 | - |
| BiSAGE | 80.8±5.9 | - | - | - | 64.9±1.2 | - |
| BiGAT | 58.1±4.4 | - | - | - | 67.1±1.4 | - |
| DGCN | 67.3±4.3 | 71.7±7.4 | 65.5±4.7 | 81.3±1.4 | 66.3±2.0 | **90.4±5.6** |
| Digraph | 66.8±6.2 | 64.9±8.1 | 59.6±3.8 | 79.4±1.8 | 62.6±2.2 | 82.0±3.1 |
| DiGraphIB | 64.4±9.0 | 64.9±13.7 | 64.1±7.0 | 79.3± 1.2 | 61.1±1.7 | 64.1±7.0 |
| KNN | 44.6±8.0 | 61.4±7.3 | 48.4±5.8 | 19.4±0.8 | 20.2±0.4 | 59.2±4.0 |
| MagNet | **84.3±7.0** | 83.3±6.1 | **85.7±3.2** | 79.8±2.5 | **67.5±1.8** | 87.6 ±2.9 |
| Best $q$ | 0.25 | 0.15 | 0.05 | 0.0 | 0.0 | 0.15 |

Table 9: Link prediction accuracy (%) with noisy labels. The best results are in **bold** and the second best are underlined.

| | Direction prediction | | | | | Existence prediction | | | | |
| --- | --- | --- | --- | --- | --- | --- | --- | --- | --- | --- |
| | Cornell | Texas | Wisconsin | Cora-ML | CiteSeer | Cornell | Texas | Wisconsin | Cora-ML | CiteSeer |
| ChebNet | 71.0±5.5 | 66.8±6.9 | 67.5±4.5 | 72.7±1.5 | 68.0±1.6 | 80.1±2.3 | 81.7±2.7 | 82.5±1.9 | 80.0±0.6 | 77.4±0.4 |
| GCN | 56.2±8.7 | 69.8±4.9 | 71.0±4.0 | 79.8±1.1 | 68.9±2.8 | 75.1±1.4 | 76.1±3.0 | 75.1±1.9 | 81.6±0.5 | 76.9±0.5 |
| APPNP | 69.5±9.0 | 76.8±5.1 | 75.1±3.5 | 83.7±0.7 | 77.9±1.6 | 74.9±1.5 | 76.4±2.5 | 75.7±2.2 | 82.5±0.6 | 78.6±0.7 |
| SAGE | 75.2±11.0 | 69.8±5.9 | 72.0±3.5 | 68.2±0.8 | 68.7±1.5 | 79.8±2.4 | 75.2±3.1 | 77.3±2.9 | 75.0±0.0 | 74.1±1.0 |
| GIN | 69.3±6.0 | 76.1±4.5 | 74.8±3.7 | 83.2±0.9 | 76.3±1.4 | 74.5±2.1 | 77.5±3.8 | 76.2±1.9 | 82.5±0.7 | 77.9±0.7 |
| GAT | 67.9±11.1 | 50.0±2.0 | 53.2±2.6 | 50.0±0.1 | 50.6±0.5 | 77.9±3.2 | 74.9±0.3 | 74.6±0.0 | 75.0±0.0 | 75.0±0.0 |
| DGCN | 80.7±6.3 | 72.5±8.0 | 74.5±7.2 | 79.6±1.5 | 78.5±2.3 | 80.0±3.9 | 82.3±3.1 | **82.8±2.0** | 82.1±0.5 | 81.2±0.4 |
| DiGraph | 79.3±1.9 | 79.8±3.0 | 82.3±4.9 | 80.8±1.1 | 81.0±1.1 | 80.6±2.5 | 82.8±2.5 | **82.8±2.6** | 81.8±0.5 | **82.2±0.6** |
| DiGraphIB | 79.8±4.8 | **81.1±2.5** | 82.0±4.9 | 83.4±1.1 | 82.5±1.3 | 80.5±3.6 | **83.6±2.6** | 82.4±2.2 | 82.2±0.5 | 81.0±0.5 |
| MagNet | **82.9±3.5** | 80.9±4.2 | **83.3±3.0** | **86.5±0.7** | **84.8±1.2** | **81.1±3.3** | **83.6±2.7** | **82.8±2.2** | **82.7±0.7** | 79.9±0.6 |
| Best q | 0.20 | 0.15 | 0.10 | 0.20 | 0.15 | 0.25 | 0.10 | 0.05 | 0.05 | 0.05 |

Table 10: Link prediction accuracy based on three classes labels(%) with noisy labels. The best results are in **bold** and the second best are underlined.

| | Three classes link prediction | | | | | Direction prediction by three classes training | | | | |
| --- | --- | --- | --- | --- | --- | --- | --- | --- | --- | --- |
| | Cornell | Texas | Wisconsin | Cora-ML | CiteSeer | Cornell | Texas | Wisconsin | Cora-ML | CiteSeer |
| ChebNet | 60.0±2.3 | 64.8±2.4 | 65.9±2.8 | 64.8±0.7 | 58.0±0.9 | 70.7±2.8 | 58.9±4.7 | 67.0±4.1 | 72.6±1.3 | 67.8±1.8 |
| GCN | 51.9±3.2 | 54.1±3.0 | 51.7±2.6 | 66.9±0.5 | 54.6±1.2 | 56.0±8.1 | 69.8±5.4 | 68.0±3.2 | 79.7±1.1 | 68.8±2.2 |
| APPNP | 55.4±4.9 | 56.7±2.7 | 53.7±3.2 | 67.8±0.7 | 57.8±1.2 | 76.0±6.6 | 76.4±4.0 | 72.5±3.9 | 82.5±0.7 | 78.2±1.7 |
| SAGE | 62.1±4.2 | 52.2±2.6 | 56.4±3.9 | 50.0±0.0 | 48.9±1.1 | 74.8±6.7 | 69.5±4.2 | 71.4±3.6 | 68.2±0.9 | 68.7±1.8 |
| GIN | 52.3±5.2 | 57.6±2.4 | 54.9±3.5 | **68.0±0.8** | 56.8±1.3 | 69.5±5.5 | 76.6±4.3 | 74.2±4.0 | 83.2±1.0 | 76.3±1.3 |
| GAT | 57.3±6.4 | 50.0±0.0 | 49.9±0.8 | 50.0±0.0 | 50.0±0.1 | 69.0±6.8 | 50.5±1.7 | 51.4±2.4 | 50.1±0.1 | 50.2±0.6 |
| DGCN | 63.2±4.7 | 64.9±2.8 | 65.9±2.7 | 67.2±0.6 | 63.7±0.7 | 76.7±4.4 | 63.4±8.9 | 68.1±6.7 | 79.4±1.4 | 78.6±2.2 |
| DiGraph | 63.7±3.7 | 66.6±3.3 | **67.2±2.2** | 66.4±0.6 | **64.7±0.7** | 79.5±1.9 | 80.0±3.8 | **82.8±5.3** | 80.7±1.0 | 79.9±1.2 |
| DiGraphIB | 62.0±5.3 | **67.4±2.5** | 66.0±1.5 | 66.0±0.6 | 62.0±0.9 | 81.4±2.8 | 79.8±3.1 | **82.8±5.2** | 83.2±1.1 | 82.5±1.6 |
| MagNet | **64.9±5.1** | 67.2±3.9 | 65.8±2.7 | 67.4±1.0 | 61.0±1.3 | **83.6±4.2** | **80.5±2.1** | 81.0±5.3 | **85.9±1.1** | **84.7±1.1** |
| Best q | 0.20 | 0.10 | 0.05 | 0.05 | 0.05 | 0.20 | 0.10 | 0.05 | 0.05 | 0.05 |

Table 11: Existence prediction(%) with noiseless labels. The best results are in **bold** and the second best are underlined.

| | Cornell | Texas | Wisconsin | Cora-ML | CiteSeer | WikiCS | Chameleon | Squirrel |
|---|---|---|---|---|---|---|---|---|
| ChebNet | 68.6±5.1 | 67.7±9.9 | 70.1±5.6 | 71.2±0.8 | 66.0±1.6 | 78.4±0.3 | 88.7±0.3 | 90.4±0.2 |
| GCN-D | 56.7±10.4 | 66.1±7.5 | 62.9±6.0 | 75.5±1.1 | 64.0±1.8 | 78.3±0.3 | 90.1±0.3 | 92.0±0.2 |
| APPNP-D | 65.2±9.0 | 72.1±6.9 | 71.5±4.0 | 78.6±0.7 | 71.0±0.8 | 80.6±0.3 | 90.4±0.4 | **91.8±0.2** |
| SAGE-D | 71.2±7.7 | 66.6±7.2 | 70.5±5.5 | 70.1±1.4 | 64.0±1.6 | 62.2±0.3 | 86.1±0.6 | 83.7±0.2 |
| GIN-D | 63.8±7.1 | 72.1±5.7 | 70.1±3.6 | 78.3±1.0 | 70.1±0.9 | 80.5±0.3 | 90.4±0.4 | 92.1±0.1 |
| GAT-D | 62.6±9.9 | 50.0±1.8 | 50.9±1.6 | 50.0±0.1 | 50.2±0.5 | 50.2±0.3 | 50.1±0.2 | 58.8±13.4 |
| BiGCN | 73.7±5.4 | - | - | - | 75.8±1.2 | - | - | - |
| BiSAGE | **79.0±6.7** | - | - | - | 72.1±1.9 | - | - | - |
| BiGAT | 74.7±6.9 | - | - | - | 50.4±0.5 | - | - | - |
| DGCN | 73.2±5.3 | 67.1±9.8 | 71.8±4.5 | 74.0±1.0 | 73.4±1.2 | 80.7±0.3 | 89.1±0.4 | 91.5±0.2 |
| DiGraph | 71.6±5.3 | 84.2±3.8 | 79.4±3.3 | 75.7±1.1 | 74.0±1.3 | 76.8±0.3 | 89.3±0.4 | 91.4±0.1 |
| DiGraphIB | 73.4±4.4 | **85.1±5.6** | 77.9±3.8 | 76.0±1.0 | 74.3±2.0 | 76.9±0.4 | 89.3±0.5 | 90.8±0.1 |
| MagNet | 76.2±4.4 | 84.9±3.9 | **81.7±2.2** | **81.1±0.7** | **80.7±0.6** | **84.2±0.2** | **91.1±0.5** | 91.6±0.2 |
| Best $q$ | 0.20 | 0.10 | 0.05 | 0.15 | 0.05 | 0.10 | 0.10 | 0.10 |

Table 12: Direction prediction(%) with noiseless labels. The best results are in **bold** and the second best are underlined.

| | Cornell | Texas | Wisconsin | Cora-ML | CiteSeer | WikiCS | Chameleon | Squirrel |
|---|---|---|---|---|---|---|---|---|
| ChebNet | 74.1±5.6 | 72.3±10.0 | 69.9±6.2 | 73.3±1.2 | 69.2±2.1 | 71.1±0.3 | 94.6±0.2 | 95.3±0.2 |
| GCN-D | 54.4±8.8 | 76.7±6.3 | 73.8±4.2 | 80.8±1.1 | 70.8±2.3 | 78.4±0.2 | 97.2±0.2 | 97.2±0.1 |
| APPNP-D | 73.6±6.6 | 83.6±4.3 | 80.8±4.5 | 85.6±0.8 | 81.0±1.8 | 82.9±0.2 | 97.6±0.2 | 98.1±0.1 |
| SAGE-D | 77.0±5.5 | 77.7±6.5 | 76.4±3.8 | 69.3±0.5 | 70.1±1.6 | 56.0±0.2 | 94.4±0.3 | 93.6±1.8 |
| GIN-D | 69.4±6.6 | 84.7±4.5 | 80.6±3.8 | 84.5±0.9 | 78.5±1.4 | 82.9±0.1 | 97.6±0.2 | 98.0±0.1 |
| GAT-D | 71.8±10.1 | 51.1±1.8 | 52.2±2.0 | 50.1±0.2 | 50.7±0.5 | 50.2±0.4 | 50.5±1.3 | 68.6±16.8 |
| BiGCN | 87.0±4.4 | - | - | - | 83.9±1.2 | - | - | - |
| BiSAGE | 84.0±4.7 | - | - | - | 80.4±1.1 | - | - | - |
| BiGAT | 81.8±5.1 | - | - | - | 50.7±0.5 | - | - | - |
| DGCN | 82.9±5.9 | 80.8±10.8 | 76.8±8.8 | 80.3±1.5 | 81.6±2.0 | 81.6±0.3 | 96.6±0.2 | 98.0±0.1 |
| DiGraph | 83.1±4.9 | 89.0±2.8 | 87.8±4.1 | 82.0±1.0 | 84.0±1.5 | 79.6±0.2 | 97.1±0.2 | 96.9±0.1 |
| DiGraphIB | 83.7±5.6 | 89.5±3.3 | 87.8±3.9 | 84.3±1.4 | 85.1±1.4 | 83.0±0.2 | 97.6±0.2 | 97.2±0.1 |
| MagNet | **88.0±2.6** | **92.6±4.6** | **88.2±3.5** | **87.6±0.8** | **87.8±1.1** | **86.3±0.3** | 97.9±0.2 | **98.3±0.1** |
| Best $q$ | 0.15 | 0.25 | 0.10 | 0.15 | 0.15 | 0.15 | 0.15 | 0.15 |

# 8 Optimal $q$ values for synthetic data

Optimal $q$ values for synthetic graphs are shown in Tables 3, 4, 5, and 6. We observe that the optimal $q$ is smaller for node classification of cyclic DSBM graphs than the ordered and noisy cyclic DSBM graphs. For cyclic DSBM graphs, the cluster is relatively clear by checking connectivity even without direction information. But the direction is crucial for classification for the other two types of DSBM graphs. It indicates that a smaller $q$ ($q < 0.15$) is enough for node classification of directed graphs when the direction is less critical. And a larger $q$ ($q > 0.15$) is needed to encode more direction information in the phase matrix for better performance. If the cluster is evident in the symmetrized adjacency matrix, we can use $q = 0$, and MagNet will reduce to ChebNet as results of Cora-ML and CiteSeer in Table 8.

Table 13: Three classes link prediction(%) with noiseless labels. The best results are in **bold** and the second best are underlined.

| | Cornell | Texas | Wisconsin | Cora-ML | CiteSeer | WikiCS | Chameleon | Squirrel |
|---|---|---|---|---|---|---|---|---|
| ChebNet | 63.0±2.1 | 71.5±2.0 | 70.5±2.1 | 65.6±0.5 | 60.3±0.8 | 74.3±0.1 | 80.7±0.3 | 83.9±0.1 |
| GCN-D | 53.0±2.5 | 62.6±2.6 | 55.8±3.1 | 67.5±0.6 | 57.1±1.1 | 74.5±0.2 | 81.3±0.3 | 86.2±0.1 |
| APPNP-D | 61.5±3.6 | 63.3±3.3 | 57.9±4.3 | **68.6±0.7** | 60.3±1.2 | 75.8±0.2 | 81.2±0.2 | 85.9±0.1 |
| SAGE-D | 64.8±4.0 | 59.2±3.2 | 60.7±4.6 | 50.9±0.1 | 51.1±1.2 | 60.8±0.1 | 70.0±0.3 | 67.3±0.2 |
| GIN-D | 54.6±3.8 | 65.2±3.7 | 58.4±4.0 | **68.6±0.7** | 59.1±1.4 | 76.2±0.2 | 81.7±0.3 | **86.5±0.3** |
| GAT-D | 58.8±6.4 | 56.9±1.4 | 54.2±1.0 | 50.8±0.1 | 52.2±0.2 | 60.8±0.1 | 54.2±0.1 | 52.5±0.0 |
| DGCN | 65.1±6.1 | 73.6±3.6 | 71.6±1.7 | 67.9±0.5 | 66.0±0.7 | **77.6±0.1** | 80.9±0.3 | 85.4±0.1 |
| DiGraph | 66.1±4.7 | **76.4±4.0** | **72.9±2.0** | 67.2±0.7 | **67.5±0.6** | 74.4±0.2 | **83.8±0.3** | 86.4±0.2 |
| DiGraphIB | 64.5±4.1 | 76.2±4.3 | 72.4±2.6 | 66.6±0.5 | 64.4±0.6 | 71.8±0.2 | 83.4±0.2 | 85.6±0.1 |
| MagNet | **67.1±4.6** | 76.1±3.7 | 70.3±2.2 | 68.4±0.9 | 63.4±1.1 | **79.6±0.1** | **83.8±0.4** | 86.0±0.1 |
| Best $q$ | 0.25 | 0.10 | 0.05 | 0.05 | 0.10 | 0.10 | 0.10 | 0.10 |

Table 14: Direction prediction by three classes link prediction(%) with noiseless labels. The best results are in **bold** and the second best are underlined.

| | Cornell | Texas | Wisconsin | Cora-ML | CiteSeer | WikiCS | Chameleon | Squirrel |
|---|---|---|---|---|---|---|---|---|
| ChebNet | 75.6±4.9 | 61.6±5.4 | 69.7±4.1 | 73.4±1.3 | 69.4±1.5 | 71.1±0.2 | 94.6±0.2 | 95.3±0.1 |
| GCN-D | 56.6±3.0 | 77.9±7.0 | 70.9±5.1 | 80.6±1.1 | 70.3±2.1 | 78.4±0.2 | 97.2±0.2 | 97.2±0.1 |
| APPNP-D | 75.5±4.5 | 83.5±4.4 | 79.9±3.4 | 83.6±0.8 | 80.7±1.4 | 82.7±0.2 | 97.5±0.2 | 98.0±0.1 |
| SAGE-D | 77.3±4.5 | 75.0±5.4 | 75.8±5.0 | 69.2±0.6 | 69.7±1.6 | 56.0±0.3 | 94.4±0.3 | 92.8±1.3 |
| GIN-D | 71.9±4.6 | 85.6±4.1 | 80.6±3.8 | 84.4±0.8 | 78.6±2.0 | 82.9±0.2 | 97.6±0.2 | 98.1±0.1 |
| GAT-D | 67.3±10.8 | 49.7±1.6 | 52.3±1.9 | 50.0±0.1 | 50.1±0.3 | 50.2±0.5 | 50.1±0.1 | 50.0±0.0 |
| BiGCN | 83.2±4.5 | - | - | - | 83.8±1.0 | - | - | - |
| BiSAGE | 86.7±4.2 | - | - | - | 79.7±1.6 | - | - | - |
| BiGAT | 80.4±8.0 | - | - | - | 50.2±0.6 | - | - | - |
| DGCN | 79.9±6.1 | 68.0±8.9 | 77.0±4.3 | 80.1±1.1 | 81.1±2.6 | 81.6±0.3 | 96.4±0.2 | 98.0±0.1 |
| DiGraph | 85.5±4.1 | 90.4±4.0 | **87.6±4.7** | 82.0±1.0 | 83.0±1.2 | 79.6±0.2 | 97.1±0.2 | 96.9±0.1 |
| DiGraphIB | 85.2±4.7 | 89.9±3.4 | 87.5±4.2 | 84.2±1.1 | 85.2±1.3 | 82.2±0.3 | 97.1±0.2 | 96.9±0.1 |
| MagNet | **88.0±3.5** | **91.7±4.1** | 86.1±6.5 | **87.2±0.9** | **88.2±1.0** | **86.4±0.2** | 97.9±0.2 | **98.3±0.1** |
| Best $q$ | 0.25 | 0.10 | 0.05 | 0.05 | 0.10 | 0.10 | 0.10 | 0.10 |