# OpenReview forum: "MagNet: A Neural Network for Directed Graphs"
_NeurIPS.cc/2021/Conference — NeurIPS 2021 Poster_

### Official Review · Reviewer_ie35 · 2021-07-05

**Rating:** 7
**Confidence:** 4

**Summary:**

This paper considers the design of graph neural networks for directed graphs. In particular, it uses the magnetic Laplacian, a complex-valued Hermitian matrix that encodes edge directions via a phase parameter, as a matrix from which convolutional filters are formed. By defining this matrix, the authors demonstrate how it can be used in place of more traditional Laplacians in a few existing graph neural network architectures. This approach is then empirically verified on synthetic and real datasets, where the utility of the the magnetic Laplacian for encoding edge directionality is demonstrated.

**Limitations And Societal Impact:**

The societal impact is not notable compared to other graph neural network literature. Limitations on scalability are mentioned at the end, but should be discussed more in the body of the paper, as mentioned in the main review.

**Main Review:**

I will begin by discussing the strengths of this paper, then move on to the weaknesses, followed by some brief comments.

**Strengths**

1. The presentation of this paper is quite nice. It is simple, well-structured, and readable. The magnetic Laplacian is defined and explained clearly, and the convolutional networks are well-explained in the context of the existing literature. The experiments are also described well. This paper was very easy to read and understand.

2. The "MagNet" architecture is not overly dependent on a particular convolutional structure, as indicated by the authors explaining how to use the magnetic Laplacian in multiple contexts. This makes the proposed approach appealing, as it is basically a drop-in change that is appropriate for handling directed graphs.

3. The empirical validation is quite convincing. The authors were quite thorough in describing their experiments, and designed the synthetic datasets carefully to demonstrate how MagNet can fully leverage the directed information while approaches based on symmetrized matrices can't. Moreover, the results on real data, including common benchmarks, are strong. Another strength of the proposed approach is in the selection of the parameter $q$ via cross-validation. Since $q$ reflects the utility of the directional information in the dataset, it is useful itself as an output of the model for interpretation of the dataset's properties, e.g., the importance of directionality in the Cora/CiteSeer networks for link prediction, but not for classification. They also compare to a good spread of other neural approaches.

4. The magnetic Laplacian is a novel addition in this context, which I have not seen before in a paper on graph neural networks.

**Weaknesses**

1. Some reasonable approaches were not addressed in the experiments. In the introduction, it is noted that the magnetic Laplacian has been used recently for community detection and clustering. On at least some of the experiments, it seems like it would be appropriate to apply these simple approaches to compare to, especially given that the node data on the synthetic dataset is randomly sampled.

2. The description of how $q$ was selected is vague. I can only guess how it was done: this should be described in more detail, perhaps in the supplement.

3. There are no comments on complexity. Although it is mentioned that this approach is not scalable in its current form in the Limitations section, there is no discussion on this in the body of the paper. I would have liked to see a better explanation of this, perhaps via a simple description of the computational complexity. This comment also holds for the selection of the $q$ parameter as mentioned above.

4. The neural architecture itself is not particularly novel, as it is directly adapted from previous papers. However, I do not view this as much of a weakness, since the point of this paper is the use of the magnetic Laplacian as a way to encode the directed structure, as well as the tuning of the parameter $q$.

**Comments**

1. Overall thoughts: this paper does not advance the field in terms of graph neural network architectures, their training, or theoretical underpinnings. However, it does introduce an operator, namely the magnetic Laplacian, that is likely unfamiliar to many in the community. By combining graph neural networks with proper parameter selection for the magnetic Laplacian, this approach proves effective in handling data on directed graphs. For this reason, I recommend this paper for acceptance.

2. This is not an important question to address, but perhaps it could improve the presentation. A common way to describe the graph Laplacian for undirected graphs is as the product $L=BB^\top$, where $B$ is the signed incidence matrix. Given that the magnetic Laplacian is Hermitian, it admits a similar factorization for some matrix other than $B$. Does it admit a factorization in terms of a modified incidence matrix?

**Editorial** (no influence on decision)

1. Line 42, "informatio" is misspelled

2. Lines 104-105, "component-wise" and "componentwise" are not consistent

3. Lines 58,189, and elsewhere, "link-prediction" and "link prediction" are not consistent

4. The plot in Figure 2 isn't very attractive. Perhaps the legend should be separated from all the plots to emphasize that it is shared, as well as to make it more readable.

5. Some of the citations do not look very good. Please take another pass on these, with a particular focus on capitalization. Some problematic citations I noticed were [7],[9],[10],[15],[18],[23],[25],[28],[29]. There are also inconsistencies on venue citation, for instance between [1],[20], and [36], or in capitalization between [10] and [11]. I would suggest using the `@string` macro in BibTeX to ensure consistent venue naming, e.g.,

```
@string{iclr={International Conference on Learning Representations}}

@article{
title={Absurdly Deep Convolutional Neural Networks},
author={A. Authors},
journal=iclr,
}
```

**After Rebuttal**

Thanks to the authors for considering my comments as well as the comments of the other authors. I am not sure if the paper is in the top 50% of accepted papers, but I still believe that it should be accepted. Thus, I am keeping my score the same. The authors have done a good job with this work.

**Time Spent Reviewing:**

4

---

> ### Author Response · Authors · 2021-08-10
> **Initial response to Reviewer ie35**
>
> Thank you for the review. Please see below for our responses to your questions, concerns, and comments.
>
> **Reviewer:** Some reasonable approaches were not addressed in the experiments. In the introduction, it is noted that the magnetic Laplacian has been used recently for community detection and clustering.  On at least some of the experiments, it seems like it would be appropriate to apply these simple approaches to compare to, especially given that the node data on the synthetic dataset is randomly sample
>
> **Response:** Thank you for this suggestion. Based on [17], we have run a magnetic Laplacian, spectral-clustering based approach (with a KNN back end) for node classification on both our noisy cyclic DSBM dataset and on our real-world datasets. On the DSBM datasets, the spectral KNN classifier performs extremely well and exceeds the performance of MagNet. This is perhaps unsurprising since it is  known that spectral clustering style methods  generally perform well on stochastic block models. However, we note that in our main experiments with MagNet, we used random noise for our initial node features $X$. If we rerun these experiments using the lead eigenvector of the magnetic Laplacian for the initial node features, then the performance of MagNet improves significantly and exceeds that of spectral clustering.
>
> For real datasets, spectral clustering performs poorly, with the following testing accuracy (\%) for node classification: Cornell: 44.6, Texas: 61.4, Wisconsin: 48.4, Cora: 19.4, Citeseer: 20.2, Telegram: 59.2. In comparison, MagNet achieves: Cornell: 84.3, Texas: 83.3, Wisconsin: 85.7, Cora: 79.8, Citeseer: 67.5, Telegram: 87.6. The poor performance of spectral clustering may be caused by the inability of spectral clustering to leverage the important information which is contained in the node features of these datasets.
>
> **Reviewer:** The description of how $q$ was selected is vague. I can only guess how it was done: this should be described in more detail, perhaps in the supplement.
>
> **Response:** q was selected by cross validation from possible choices of 0, 0.05, 0.1, 0.15, 0.2, and 0.25. We apologize for this accidental omission and will clarify this upon revision.
>
> **Reviewer:** There are no comments on complexity. Although it is mentioned that this approach is not scalable in its current form in the Limitations section, there is no discussion on this in the body of the paper. I would have liked to see a better explanation of this, perhaps via a simple description of the computational complexity. This comment also holds for the selection of the $q$ parameter as mentioned above.
>
> **Response:** The complexity of the proposed algorithm is comparable to standard GCN algorithms (although our method does differ by a factor of four because of the computational complexity of complex-valued multiplication). Moreover, the various techniques that can increase efficiency of standard GCNs can also be extended our constructions. We will add some discussion of this point upon revision.
>
> **Reviewer:** A common way to describe the graph Laplacian for undirected graphs is as the product $L=BB^T$, where $B$ is the signed incidence matrix. Given that the magnetic Laplacian is Hermitian, it admits a similar factorization for some matrix other than $B$. Does it admit a factorization in terms of a modified incidence matrix?
>
> **Response:** We had not thought about this previously, but it turns out the answer is "yes." One may factor the unnormalized magnetic Laplacian as $L_U^{(q)}=BB^{\dagger}$ where $B$ is a modified incidence matrix whose nonzero entries are given by $B_{j,\ell}=\frac{1}{\sqrt{2}}e^{iq/2}$ if vertex $j$ is the "source" of link $\ell$ and $B_{j,\ell}=\frac{-1}{\sqrt{2}}e^{-iq/2}$ if vertex $j$ is the "target" of link $\ell$. Upon revision, we will discuss this point in the main text and add a proof to the supplement.

---

### Official Review · Reviewer_YVEh · 2021-07-07

**Rating:** 6
**Confidence:** 4

**Summary:**

This paper proposes to use magnetic Laplacian within GNNs to better capture the direction information. They introduce a spectral GNN based on this and show the performance on some node and link prediction tasks.

**Limitations And Societal Impact:**

Yes

**Main Review:**

The paper is well-written and easy to follow. It also has a good justification of why their method works, and in practice they show that they are getting good results on some node and link prediction benchmarks. The idea is not new in graph signal processing community but it seems it is the first time that is being incorporated into GNNs as an inductive bias. There are three main concerns/questions that I have:

1- Sparse representation of graphs is very important as many real-world graph problems deal with huge graphs. However it seems that the exponential of phase matrix is dense because even if there is no connection between two nodes the exp(phase) will be 1. Can you comment on this?

2 - One simple baseline for directed graphs can be decoupling the incoming edges from outgoing edges for each node, learn two disentangled representations, and then aggregate them. Any thoughts how this simple baseline would perform compared to  your model? Also, approaches such as Graph diffusion convolution (GDC) [1], or Graphormer [2] should also be able to capture direction information using diffusion and structural self-attention, respectively. I wonder how such models would perform compared to yours.

3- Also I have two concerns regarding the performance and scalability of the proposed models. Especifically, it seems that all your evaluations are performed on transductive taks and it is not clear how your method would work on inductive tasks. Moreover, the benchmarks that you have chosen are fairly small with maximum of  3,312 nodes for node prediction and 11,701 nodes for link prediction tasks. I recommend to evaluate on more realistic directed benchmarks such as ogbn-papers100M, ogbn-arxiv, ogbn-mag, ogbl-citation2, ogbl-wikikg2. (Thanks for explicitly mentioning this in limitations section: "is not scalable to large graphs in
its current form (although this may be addressed in future work)."

4- Finally, I couldn't find information about how you would treat edge features in your framework if available.


[1] https://proceedings.neurips.cc/paper/2019/hash/23c894276a2c5a16470e6a31f4618d73-Abstract.html

[2] https://arxiv.org/abs/2106.05234

After receiving the author responses and also reading the reviews of the other reviewers, I decided to increase the score. However, I still think showing results for inductive tasks and tasks with edge features can make the paper stronger.

**Time Spent Reviewing:**

2

---

> ### Author Response · Authors · 2021-08-10
> **Initial response to Reviewer YVEh**
>
> Thank you for the review. Please see below for our responses to your questions, concerns, and comments.
>
> **Reviewer:** Sparse representation of graphs is very important as many real-world graph problems deal with huge graphs. However it seems that the exponential of phase matrix is dense because even if there is no connection between two nodes the exp(phase) will be 1. Can you comment on this?
>
> **Response:** The reviewer correctly notes that the phase matrix is dense in theory. However, since the Hermitian adjacency matrix is constructed via pointwise multiplication between the symmetrized adjacency matrix and the phase matrix, it is only necessary to compute (and store) the phase matrix for entries $(u,v)$ where the symmetrized adjacency matrix is nonzero. Therefore, in practice, we may treat the phase matrix as having the same sparsity as the symmetrized adjacency matrix.
>
> **Reviewer:** One simple baseline for directed graphs can be decoupling the incoming edges from outgoing edges for each node, learn two disentangled representations, and then aggregate them. Any thoughts how this simple baseline would perform compared to your model? Also, approaches such as Graph diffusion convolution (GDC) [1], or Graphormer [2] should also be able to capture direction information using diffusion and structural self-attention, respectively. I wonder how such models would perform compared to yours.
>
> **Response:** In order to respond to this question, we developed decoupled incoming/outgoing representations for GCN, GAT, and SAGE, and tested these methods on two real-world data sets, Cornell and Citeseer, as well as on our noisy cyclic directed stochastic block model (DSBM). For Cornell and Citeseer, we tested on node classification as well as three types of link prediction (i.e., eight tasks), and for the noisy cyclic DSBM data, we utilized four parameter settings (i.e., 4 node classification tasks). Therefore, all together, there are six node classification tasks and six link predictions tasks. MagNet was the top performer on 11/12 of these tasks and was second on the other task. This illustrates a clear benefit to using MagNet over this type of baseline approach.
>
> With respect to GDC, as we understand it, the algorithm is a pre-processing step and so to analyze it we would need to pre-process the data using GDC and then re-run all the algorithms on the pre-processed data. Having said that, we point out that we did compare to APPNP, which is based off a graph diffusion, specifically personalized page rank.
>
> Finally, with respect to Graphormer, this paper was not posted on arXiv until June 9, 2021, which was after the submission deadline for NeurIPS.
>
> **Reviewer:** Also I have two concerns regarding the performance and scalability of the proposed models. Especifically, it seems that all your evaluations are performed on transductive tasks and it is not clear how your method would work on inductive tasks. Moreover, the benchmarks that you have chosen are fairly small with maximum of 3,312 nodes for node prediction and 11,701 nodes for link prediction tasks. I recommend to evaluate on more realistic directed benchmarks such as ogbn-papers100M, ogbn-arxiv, ogbn-mag, ogbl-citation2, ogbl-wikikg2. (Thanks for explicitly mentioning this in limitations section: ”is not scalable to large graphsin its current form (although this may be addressed in future work).”
>
> **Response:** We focused on transductive tasks to allow for straightforward comparison to GCN and its variants, as well as other papers on directed graph neural networks (since these papers also primarily focus on transductive tasks). For inductive tasks such as graph classification/regression, the back end of ChebNet and GCN can be easily modified to obtain graph level features by using a global node aggregation function such as node summation (note, this is a common approach). The same method can be applied to MagNet. We also point out that many directed graph data sets are for transductive tasks, and there is a relative lack of directed graph data sets for inductive tasks.
>
> With respect to the larger data sets, we agree that testing on these data sets would be a valuable contribution. However, the goal of this paper is to introduce the MagNet architecture and to illustrate its potential for improving the state-of-the-art in GNNs on directed graph tasks. We also point out, that any approaches for scaling GCN on undirected graphs will be applicable to MagNet on directed graphs.
>
> **Reviewer:** Finally, I couldn’t find information about how you would treat edge features in your frame-work if available.
>
> **Response:** The focus of our paper is graphs with node features, not graphs with edge features. That said, there are certain standard tricks that could be used to apply our network to graphs with edge features. For example, one could create an additional node ``in the middle" of each directed edge and use the edge features to define node features for these new nodes. (Direction would be preserved in the sense that if $(u,v)$ was a directed edge in the original graph, then $(u,v_{new})$ and $(v_{new},v)$ would be directed edges in the new graph.)

---

### Official Review · Reviewer_3ZNa · 2021-07-15

**Rating:** 7
**Confidence:** 4

**Summary:**

This work focus on modeling directed graph in the spectral domain, which is not handled well by current spatial and spectral methods. The major contribution is using the Hermitian matrix to encode direction information.

**Limitations And Societal Impact:**

- describe how the proposed method distinguishes forward, backward, bi-direction, and no link.
- discuss and study the efficiency problem: how much will the additional phase matrix and its multiplication take?
- present the implementation of multiplication between complex matrices

**Main Review:**


This paper proposed a graph neural network for directed graphs. To encode direction, the Hermitian matrix is used as a Laplacian substitute in normal graph convolution.

### 1 Motivation
- Directed graph is not well handled by the current graph neural network, and the author clearly presents the pros and cons of spectral and spatial methods, showing a new approach is needed.
- The example in the introduction shows a suitable potential to address the directionality. However, it is not clear if the Hermitian matrix can distinguish forward, backward, bi-direction, and no link.
- In the spatial method, what is the benefit to consider a neighbor even it is not connected through a directed link?

### 2 Presentation- Overall, the presentation is good, but there exist several weaknesses.
- Sections 2 and 3 present a workflow that is very similar to the most popular graph convolution. Obviously, it is better to focus on the proposed component.
- several related works are missing such as ICML 21 Directed Graph Embeddings in Pseudo-Riemannian Manifolds, and ICLR 20 DIMENet

### 3 Technical analysis- The paper proposes Hermitian matrices as a substitute for graph Laplacian.
- The major concern is that how the Hermitian matrix can distinguish directions. In the definition of phase matrix in line 103, how to tell (1) no link between u and v, from (2) bi-directional link between u and v ? Do they both equal to zero?
- The phase matrix is as large as the adjacency matrix, there may be an efficiency issue when the graph is large.

### 4 Experiment- The proposed method has been evaluated on both synthetic and real-world graph datasets.
- The results of link prediction well support the proposed method, but the performance on the node classification task is not significantly better than the baselines.
- Since there might be an efficiency issue, efficiency evaluation is expected.

### After rebuttal
The author well cleared my concerns and the score was updated.


**Time Spent Reviewing:**

5

---

> ### Author Response · Authors · 2021-08-10
> **Initial response to Reviewer 3ZNa**
>
> Thank you for the review. Please see below for our responses to your questions, concerns, and comments.
>
> **Reviewer:** The example in the introduction shows a suitable potential to address the directionality. However, it is not clear if the Hermitian matrix can distinguish forward, backward, bi-direction, and no link.
>
> **Response:** It can distinguish these four cases as follows. The Hermitian adjacency matrix is obtained by pointwise multiplying the phase matrix and the symmetrized adjacency matrix. Therefore, if there is no-link, we will have $H^q(u,v)=0.$ If there is a bi-directional link, then the phase matrix $\Theta$ will be zero and so $H^{q}(u,v)$ will be non-zero and real-valued. Lastly, in the case of forward and backward links, the Hermitian adjacency will be complex valued, and changing a forward link to a backward link will correspond to complex conjugation.
>
> **Reviewer:** In the spatial method, what is the benefit to consider a neighbor even it is not connected through a directed link?
>
> **Response:** Here are two examples to illustrate the importance. First, consider citation networks in which each node is a paper, a directed edge $(a,b) \in E$ means paper $a$ cites paper $b$, so that $b$ is a neighbor of $a$ but $a$ is not a neighbor of $b$. If you want to infer something about paper $v$, then there is important information in the papers that $v$ cites, i.e., the the neighbors of $v$. However, there is also important information in the papers that cite $v$, which are not directed neighbors of $v$ using the standard definition of neighborhoods because they correspond to edges of the form $(b,v)$ rather than $(v,b)$. Second, consider a directed social network such as Twitter. Let nodes be Twitter accounts and a directed edge $(a,b) \in E$ means that account $a$ mentions account $b$ (using the @ functionality). Similar to the citation network example, if you want to infer something about account $v$, there is important information to be gathered both from other accounts that $v$ mentions (neighbors of $v$), and accounts that mention $v$ (not neighbors of $v$ in a directed graph). Also note that in both of these examples, it may be important to distinguish between these two types of links and that   Hermitian adjacency matrices are well equipped to do this.
>
> **Reviewer:** Sections 2 and 3 present a workflow that is very similar to the most popular graph convolution. Obviously, it is better to focus on the proposed component.
>
> **Response:** Section 2 describes Hermitian adjacency matrices and the magnetic Laplacian. These types of matrices are not commonly used in the machine learning/graph neural network community, and therefore we did not assume the reader has knowledge of them. For section 3, we wanted to make the paper as self-contained as possible and also establish notation. Additionally, this section illustrates that the graph convolution constructions still work for Hermitian, complex valued matrices, which may not be obvious to some readers.
>
> **Reviewer:** several related works are missing such as ICML 21 Directed Graph Embeddings in Pseudo-Riemannian Manifolds, and ICLR 20 DIMENet.
>
> **Response:** Thank you very much for pointing this out. We will add a brief discussion of these papers to the Related Work section. Moreover, we feel that this will strengthen the paper by showing that directed graphs are an increasingly important area of research in data science and that they are relevant to potentially surprising areas of application such as predicting molecular energies as in DIMENet.
>
> **Reviewer:** The major concern is that how the Hermitian matrix can distinguish directions. In the definition of phase matrix in line 103, how to tell (1) no link between u and v, from (2) bi-directional link between u and v? Do they both equal to zero?
>
> **Response:** As mentioned above, the Hermitian adjacency matrix will be zero in the case of no link and a non-zero real number in the case of a bi-directional link.
>
> **Reviewer:** The phase matrix is as large as the adjacency matrix, there may be an efficiency issue when the graph is large
>
> **Response:** For most real-world datasets, the symmetrized adjacency matrix will be sparse. Moreover, we note that since the Hermitian adjacency matrix is constructed via pointwise multiplication between the symmetrized adjacency matrix and the phase matrix, it is only necessary to compute the phase matrix for entries $(u,v)$ where the symmetrized adjacency is nonzero. Thus, the efficiency of the proposed algorithm is comparable to standard GCN algorithms, and can leverage any existing/future developments that increase efficiency of standard GCNs (although the computational complexity our method does differ by a factor of four because of the computational complexity of complex-valued multiplication); for example, the recent paper "GNNAutoScale: Scalable and Expressive Graph Neural Networks via Historical Embeddings" proposes a way to scale GCN, and could apply to MagNet to as well.
>
> **Reviewer:** The results of link prediction well support the proposed method, but the performance on the node classification task is not significantly better than the baselines.
>
> **Response:** Respectfully, we feel that the improved performance of our network for node classification is significant. For node classification, MagNet is the top method on three out of six tasks and comes in second on another two. No other method is the top method on more than one task. SAGE is seemingly the next most competitive method (one top performance, three second best performances), but it completely fails on the telegram data. We also note the methods designed specifically for directed graphs perform significantly worse than MagNet on the Cornell, Texas, and Wisconsin data sets. To make this analysis more quantitative, we computed the absolute difference of the classification accuracy of each method from the classification accuracy of the top performing method (in percentage points) on each data set, and averaged over the six data sets reported in Table 1 (thus, the lower the score the better, and in particular, a method with a score of zero indicates the method is the top performing method on each data set). MagNet achieved a score of 1.1 percent, whereas the next best were ChebNet and SAGE with 6.9 and 7.4 respectively. This is a significant gap. Across all methods the scores were: GCN: 19.2, Cheb: 6.9, APPNP: 18.8, SAGE: 7.4, GAT: 16.4, GIN: 18.2, DGCN: 8.5, Digraph: 15.7, DigraphIB: 16.4, Magnet: 1.1
>
> **Reviewer:** Present the implementation of multiplication between complex matrices
>
> **Response:** Since complex tensors are still in beta in PyTorch, we did not use them, and instead we stored any complex tensor as two real tensors (one for the real part, one for the imaginary part), and carried out complex multiplication using the standard formula: $(a + ib)(c + id) = (ac - bd) + i(bc + ad)$ (note, $a,b,c,d$ can be real numbers or real matrices).

---

### Decision · Program_Chairs · 2021-09-27

**Decision:**

Accept (Poster)

**Comment:**

The paper proposes a new GNN for directed graphs using magnetic Laplacian. Specifically, the authors use a complex operator encoding edge direction via phase. This is an interesting approach, which was appreciated by the reviewers. The authors provided detailed responses that addressed the reviewers' comments in a satisfactory manner. The AC recommends acceptance.